# Evolutionary Dynamics of Gig Economy Labor Strategies under Technology, Policy and Market Influence

**Kevin Hu** [1] and **Feng Fu** [1,2,*]

1   Department of Mathematics, Dartmouth College, Hanover, NH 03755, USA; kevn.hu@gmail.com
2   Department of Biomedical Data Science, Geisel School of Medicine at Dartmouth, Lebanon, NH 03756, USA
*   Correspondence: fufeng@gmail.com; Tel.: +1-603-646-2293

**Abstract:** The emergence of the modern gig economy introduces a new set of employment considerations for firms and laborers that include various trade-offs. With a game-theoretical approach, we examine the influences of technology, policy and markets on firm and worker preferences for gig labor. Theoretically, we present new conceptual extensions to the replicator equation and model oscillating dynamics in two-player asymmetric bi-matrix games with time-evolving environments, introducing concepts of the attractor arc, trapping zone and escape. While canonical applications of evolutionary game theory focus on the evolutionary stable strategy, our model assumes that the system exhibits oscillatory dynamics and can persist for long temporal intervals in a pseudo-stable state. We demonstrate how changing market conditions result in distinct evolutionary patterns across labor economies. Informing tensions regarding the future of this new employment category, we present a novel payoff framework to analyze the role of technology on the growth of the gig economy. Regarding governance, we explore regulatory implications within the gig economy, demonstrating how intervals of lenient and strict policy alter firm and worker sensitivities between gig and employee labor strategies. Finally, we establish an aggregate economic framework to explain how technology, policy and market environments engage in an interlocking dance, a balancing act, to sustain the observable co-existence of gig and employee labor strategies.

**Keywords:** evolutionary game theory; gig worker; evolutionary economics; social learning; oscillatory dynamics

**JEL Classification:** C73; E27

## 1. Introduction

With economic prevalence that extends to the labor markets of the early Roman Empire [1], the concept of contract work has existed for millennia, manifesting in different forms across societies and temporal interludes [2]. In recent decades, contract or 'gig' work has emerged as a commanding employment category in the United States, having captured more than one third of the labor market by 2018 [3]. At the cornerstone of this development are online labor marketplaces that facilitate the exchange of talent and capital between firms and workers, effectively decreasing hiring frictions and increasing labor liquidity [4–6]. The result of this infrastructure furtherance takes form in a novel, complementary contract-based labor market monikered the sharing, collaborative or gig economy [7–10].

Specific to the rise of gig work in the digital era, the modern gig economy enables firms to digitally outsource tasks and processes to remote workforces and match independent skill sets to specific labor needs [11]. For instance, ride-share companies such as Lyft and Uber leverage contractual gig drivers in their businesses, ultimately re-engineering cheaper, on-demand product offerings [12,13]. There is, however, a trade-off; the commitment to cheaper pricing with gig operators may come at the expense of product and service quality [14]. On the labor supply side, autonomy, self governance and overall increased

flexibility form the gravitational kernel that captivates new workers and persuades them to participate in the gig economy [15,16]. However, gig workers lack the income stability and labor protections such as union rights and insurance benefits conferred with employee status [17]. For each labor economy, there exists a crossroads of competing considerations for firms and laborers regarding their labor decisions.

Beyond the firm and individual, there are several macro factors at play. The dynamics of firm and worker labor preferences are saddled at the nexus between *market conditions, technology and policy*. Noting select periods of American economic history from the last century, we see a pattern wherein which the importance and popularity of contract work fluctuate as a result of several economic factors. Notably, during the post Great Depression and World War II period, workers sought out an auxiliary arrangement, a reconstitution of work and enterprise, in a pursuit of autonomy and stability [18]. Over the last century, this campaign for autonomy, not contemporary digital applications, set the foundation for the modern gig economy [18]. Regarding recent economic cycles, present-day gig workers recognize that the structural forces of economic recessions restrict their autonomy; when demand for work declines, gig laborers remain persistently available to compete for limited contracts, thereby disqualifying any scheduling flexibility [15]. A market–labor pattern emerges across history and informs us on how evolving market cycles shape the labor landscape.

There is also compelling evidence to believe that technological advancements may engender the future growth or stagnation of the gig economy. On the one hand, there is an expectation that the gig economy will continue to grow with the introduction of new sharing platforms and businesses [10,19]. On the other hand, there exists a growing accord in scholarship that artificial intelligence (AI) will displace many human operators [4], especially those with commodity skills [20]. The rapid acceleration of AI may implicate the displacement of gig workers, for instance, the substitution of ride-sharing drivers with the introduction of autonomous vehicles [4]. A question remains as to whether these displaced workers will reenter the workforce as employees or gig workers. Seemingly, the influence of technology on the future of the gig economy depends on a constellation of co-developing technologies racing to fruition.

In recent years, there has been growing effort in studying the gig economy, which provides useful insights that address labor preferences, policy design, the role of technology and wide-ranging socioeconomic implications.

Among others, one main approach used to study the gig economy is ethnography with various statistical methods. Much has been explored regarding influences on firm and worker gig-economy incentives. Allon et al. collaborate with a ride-sharing platform to investigate behavioral and economic incentives for gig workers, noting a prioritization of an earnings goal over the number of hours worked and a willingness to work more with more hours worked [21]. Lehdonvirta explores flexibility in the gig economy, reiterating emphasis on the income-target and finds support that worker autonomy depends on a large availability of work [15]. Burtch et al. study how gig-economy platforms influence entrepreneurial activity, finding that gig platforms reduce total entrepreneurial activity as these platforms provide prospective entrepreneurs an additional stream of income [7]. Leung examines hiring in the gig economy as a learning experience, noting that firms expressed loss-aversion behaviors when responding to positive and negative hiring experiences [22]. Exploring hiring across the global gig-economy, Galperin et al. note discriminatory geographical preferences in firms' hiring preferences [23].

Academic research on the gig economy has also extensively embraced concerns in policy, technology and economics. Friedman argues that the growth of the gig economy requires new social policy as economic risks are shifted from the firm to the laborer [24]. Todoli-Signes examines the gig worker's need for protection and details regulatory concern around working hours, minimum wage, child labor bans and annual leave among other areas of apprehension [25]. Stewart and Stanford investigate five regulatory mechanisms in the gig economy such as the creation of a new independent worker category or the provision of workers' rights, reviewing the pros and cons of each framework [26]. While research

focusing on regulation and policy collectively exhibit a concern regarding the gig economy, many scholarly works on technological developments concentrate on drivers of growth for this new employment sector. In this work, we consolidate many of the aforementioned areas of research and, from game theoretical perspective, study the influence of policy, technology and market changes on firm and laborer preferences in the gig economy.

As the modern gig economy grows out of its unhampered infancy, policy makers and researchers alike are presented the question of how this market should be regulated [11,27]. Undefined ordinance allows new competitors leveraging gig work to play by different rules than industry incumbents, a result of ambiguous labor laws that enable firms to shift economic burdens onto the gig laborer [25,28,29]. In industry, some governments have mandated that firms more closely classify gig workers as employees, a decree that demands additional securities for gig laborers [30,31]. The question as to whether or how this new labor sector should be policed remains unanswered, an inquiry of apprehension we aim to inform about in the present work.

Pioneered by John von Neumann [32], the study of modern game theory anchors itself in the assumption that players make rational decisions based on the respective payoff incentives conferred with each strategy [33–35]. Although modern game theory initially focused on static theories, differential game theory, a subject introduced by Rufus Isaacs, considers the state of players with time as a continuous variable [36]. While classical game theory was developed to address questions in economics [34,37], the field of evolutionary game theory, a theoretical extension that models how populations change strategies over time [38], finds its roots in biology [33,38]. Since its inception in 1973 [33], evolutionary game theory has broadened in application beyond its early biological origins to study social interactions and population behaviors across various academic fields [38–43].

In evolutionary dynamics, the approach with the replicator equation is most notable. Originally presented by Taylor and Jonker in 1978 [44] and formally named by Schuster and Sigmund [45], the replicator equation determines the evolution of the composition of strategies in a population [46].

Using a game-theoretical approach, we investigate both firm and individual labor considerations as well as the economic influences of markets, technology and policy on labor preferences in the gig economy. In this paper, we present new conceptual extensions to the replicator equation, oscillating replicator dynamics with attractor arcs, formally, an oscillating replicator dynamics of two player asymmetric bi-matrix games with time-evolving environment. Previous studies have analyzed oscillating tragedy of the commons for evolutionary games with environmental feedback [35,47–51]. Economic behavior in the labor economy has been investigated through the lens of both evolutionary game theory and differential game theory [52–55]. For instance, Sadik-Zada derived feasible bargaining equilibria including the antagonistic and the allocation modes, based on a non-cooperative differential game model [52]. Using the framework of replicator dynamics [44,46,56], we model the evolutionary behavior of firm and laborer preferences for gig strategies. While we base our model on existing works in evolutionary dynamics, ours, to our knowledge, is the first to introduce the concept of the attractor arc, environment-actuated driven oscillation, trapping zone and escape. We discover an oscillatory fluctuation between labor strategies across market cycles as well as additional transformations resulting from various technology and policy landscapes.

In this paper, we impart three notable contributions to the field of evolutionary dynamics and existing literature on the gig economy (and more broadly non-standard work arrangements). First, we introduce a new type of game, replicator dynamics with attractor arcs. We present our model by formalizing our concepts of the attractor arc, environment-actuated driven oscillation, trapping zone and escape. While canonical applications of evolutionary game theory focus on the evolutionary stable strategy (ESS), our model assumes that the system exhibits oscillatory dynamics and can persist for long temporal intervals in a pseudo-stable state. In our theoretical extensions, we show how the attractor arc can drift around the phase space and change orientation to reflect evolving labor market composition and dynamic

strategy sensitivities. Second, we present a generalized model to study labor economies and demonstrate how market, technology and policy influence labor strategies in three distinct dynamics. Third, we establish an aggregate economic framework to explain how market, technology and policy environments engage in an interlocking dance, a balancing act, to sustain the co-existence of gig and employee labor strategies.

As detailed below, we provide researchers, policy makers and industrialists alike with a proof-of-concept evolutionary game theory approach along with an environment-dependent payoff generation framework for better understanding firm and laborer behaviors in the gig economy. Our modeling framework and the concepts including attractor arc, trapping zone and pseudo-stable state can be used to demonstrate how technology is a driver of change in the labor economy and how policy is integral to the sustainability of new systems and the protection of involved parties. The present paper aims to further comprehension of micro and macro influences on firm and laborer incentives for gig adoption in these regards.

## 2. Material and Methods

### 2.1. Overview

In this section, we derive our evolutionary model, replicator dynamics with attractor arcs, and apply the modeling analysis to theoretical payoffs. We apply replicator dynamics to model changes in firm and laborer preferences for gig labor across bear and bull markets. Thereafter, in two theoretical extensions, we detail how changes in payoffs can be applied to study technology and policy leverage in the gig economy. Finally, we establish an aggregate theory that addresses the treble of evolutionary dynamics under technology, policy and market influence and discuss implications for the labor economy.

### 2.2. Evolutionary Dynamics of Gig Economy Labor Preferences

As what follows, we detail the derivation and characteristics of our evolutionary model. First we introduce the replicator equations for $2 \times 2$ asymmetric bi-matrix games. By means of two sample bi-matrices, we analyze the phase diagrams and discuss saddle points and initial conditions. Finally, we explore oscillatory dynamics and introduce our theory on the attractor arc, trapping zones, environment-actuated driven oscillation and escape. In the following sections, we apply the model to our generated payoffs.

Replicator Equations for Asymmetric Bi-Matrix Games

In our model, we employ the replicator equation, a differential equation that determines the evolving composition of strategies in a population [46,56,57], to study gig economy labor strategies. In particular, we are interested in how firm and laborer preferences for gig labor strategies evolve across market cycles. We provide the general replicator equation where $x_i$ denotes the proportion of strategy type $i$ in the population, $\pi_i$ is the fitness of strategy type i and $\overline{\pi}$ represents the average payoff across the entire population. Fitness of a strategy type can be understood as the expected payoff for that strategy.

$$\dot{x}_i = x_i(\pi_i - \overline{\pi}) \tag{1}$$

For asymmetric bi-matrix games, replicator equations take the following form where $\dot{x}_i$ denotes the evolution for player 1 strategies and $\dot{y}_i$ denotes the evolution for player 2 strategies. In our model, player 1 is the laborer and player 2 is the firm. A and B denote the respective payoffs in matrix form for players 1 and 2. $\vec{x}$ and $\vec{y}$ denote the strategies for players 1 and 2, respectively. In vector form, the strategy set for laborers is represented as $\vec{x} = (x_1, x_2)^T$ and the strategy set for firms as $\vec{y} = (y_1, y_2)^T$; type 1 strategies typify gig and type 2, employee. Each strategy takes a value in the domain [0,1] and represents the probability that the strategy is selected; therefore, $x_1 + x_2 = 1$ and $y_1 + y_2 = 1$.

$$\dot{x}_i = x_i((A\vec{y})_i - \vec{x} \cdot (A\vec{y})) \tag{2}$$

$$\dot{y}_j = y_j((B\vec{x})_j - \vec{y} \cdot (B\vec{x})) \tag{3}$$

Selection intensity, denoted with $\omega \in [0,1]$, represents the frequency in which firms and laborers interact in the labor market. When firms and laborers do not interact in the labor market, the composition of employees and gig workers remains constant. When firms and laborers choose to participate in the labor market (i.e., firms hiring for and laborers seeking new employment roles), gig and employee decisions are determined based on respective payoff incentives, and the composition of employees and gig workers evolves accordingly. Firms and laborers will only interact in the labor market when they recognize evolving environments that influence existing strategy payoffs. Here, selection intensity can also be understood as the rate at which firms and laborers realize external factors that cause them to shift strategy preferences. In evolutionary game theory, this social learning process can be modeled as the Moran process [46,58]. The Moran process proceeds as a stochastic process where one labor contract ends and a new labor contract arises to reflect the latest firm and laborer preferences, which are modeled by the fitness of each strategy type. In our model, $\omega$ constitutes the rate of change for strategy densities in firm and laborer populations. For $\omega = 0$, the fitness of the strategy type is 0 as the player does not interact in the labor market, and the rate of change for gig-employee strategy densities is 0. When $\omega = 1$, the fitness $f$ equates to the payoff $\pi$ for the strategy type, and firms and laborers engage in the labor market at the maximum cadence. We have

$$f = 1 - \omega + \omega\pi \tag{4}$$

Since each player's strategy set sums to 1, we can mathematically represent our model with just $x_1$ and $y_1$. For $2 \times 2$ bi-matrix games incorporating selection intensity, replicator equations can be represented in the following form:

$$\dot{x}_1 = \omega x_1(1 - x_1)((A\vec{y})_1 - (A\vec{y})_2) \tag{5}$$

$$\dot{y}_1 = \omega y_1(1 - y_1)((B\vec{x})_1 - (B\vec{x})_2) \tag{6}$$

Our model involves a pair of GameStates. GameState pairs consist of a labor economy, whether that of a country or of a single firm, in a bear and bull market. Subscripts $l$ and $f$ denote laborer and firm payoffs, respectively. We append 0 and 1 to the payoff subscripts to denote bear and bull market GameStates, respectively.

<table>
<tr><td colspan="3">Bear Market GameState</td><td colspan="3">Bull Market GameState</td></tr>
<tr><td>$_{Laborer}\backslash^{Firm}$</td><td>Gig</td><td>Employee</td><td>$_{Laborer}\backslash^{Firm}$</td><td>Gig</td><td>Employee</td></tr>
<tr><td>Gig</td><td>$[a_{l0}, a_{f0}]$</td><td>$[b_{l0}, b_{f0}]$</td><td>Gig</td><td>$[a_{l1}, a_{f1}]$</td><td>$[b_{l1}, b_{f1}]$</td></tr>
<tr><td>Employee</td><td>$[c_{l0}, c_{f0}]$</td><td>$[d_{l0}, d_{f0}]$</td><td>Employee</td><td>$[c_{l1}, c_{f1}]$</td><td>$[d_{l1}, d_{f1}]$</td></tr>
</table>

We reconstitute our pair of GameState matrices, disjoining firm and laborer payoffs.

$$L_{Bear} = \begin{pmatrix} a_{l0} & b_{l0} \\ c_{l0} & d_{l0} \end{pmatrix} \quad L_{Bull} = \begin{pmatrix} a_{l1} & b_{l1} \\ c_{l1} & d_{l1} \end{pmatrix} \tag{7}$$

$$F_{Bear} = \begin{pmatrix} a_{f0} & b_{f0} \\ c_{f0} & d_{f0} \end{pmatrix}^T \quad F_{Bull} = \begin{pmatrix} a_{f1} & b_{f1} \\ c_{f1} & d_{f1} \end{pmatrix}^T \tag{8}$$

An environment coefficient, $n \in [0,1]$, represents market condition. $n = 0$ denotes the bear market and $n = 1$ denotes the bull market. $n$ can take any value between 0 and 1; for instance, $n = 0.5$ signifies that the environment is a neutral market, the midway point in a transition between bear and bull market conditions. Having introduced this environment coefficient, selection intensity $\omega$, can be understood as the social learning rate at which firms and laborers acknowledge the economic landscape and realize new payoffs for each

strategy. In our model, payoffs $A$ and $B$ are functions of time, but selection intensity $\omega$ is a constant. Applying this environment coefficient, we rephrase our firm and laborer payoffs to account for the domain of market conditions.

$$A(n) = L_{General} = \begin{pmatrix} (1-n)a_{l0} + na_{l1} & (1-n)b_{l0} + nb_{l1} \\ (1-n)c_{l0} + nc_{l1} & (1-n)d_{l0} + nd_{l1} \end{pmatrix} \tag{9}$$

$$B(n) = F_{General} = \begin{pmatrix} (1-n)a_{f0} + na_{f1} & (1-n)b_{f0} + nb_{f1} \\ (1-n)c_{f0} + nc_{f1} & (1-n)d_{f0} + nd_{f1} \end{pmatrix}^T \tag{10}$$

We apply our general firm and laborer payoffs to our replicator equations and conclude our derivation.

$$\dot{x}_1 = \omega x_1(1-x_1)(([(1-n)a_{l0} + na_{l1}]y_1 + [(1-n)b_{l0} + nb_{l1}](1-y_1)) \\ - ([(1-n)c_{l0} + nc_{l1}]y_1 + [(1-n)d_{l0} + nd_{l1}](1-y_1))) \tag{11}$$

$$\dot{y}_1 = \omega y_1(1-y_1)(([(1-n)a_{f0} + na_{f1}]x_1 + [(1-n)c_{f0} + nc_{f1}](1-x_1)) \\ - ([(1-n)b_{f0} + nb_{f1}]x_1 + [(1-n)d_{f0} + nd_{f1}](1-x_1))) \tag{12}$$

## 3. Results

### 3.1. Key Concepts and Theoretical Analysis of the Evolutionary Game Theory Model

#### 3.1.1. System Equilibria

In Appendix A.1, we solve for our evolutionary system's fixed points for the general case. For each fixed point, we examine the stability of the equilibrium by analyzing the eigenvalues of the Jacobian matrix. We find that our system has two stable fixed points at $(0,0)^*$ and $(1,1)^*$, two unstable fixed points at $(0,1)^*$ and $(1,0)^*$, and a saddle point whose position depends on firm and laborer payoff values.

#### 3.1.2. Saddle Points

We provide analysis for the saddle point with a theoretical GameState pair (we refer to Appendix A for the theoretical analysis of general cases), see Figure 1. We note that the selected theoretical payoff matrices are used in demonstrations only and have no relationships to any specific labor economy. In future sections, we generalize the model with general payoffs. For simplification purposes, we assign all mismatching strategies a payoff of 0 as no mutual labor agreement is made between firm and laborer.

**a**

| $_{Laborer}\backslash^{Firm}$ | Gig | Employee |
|---|---|---|
| Gig | [9, 3] | [0, 0] |
| Employee | [0, 0] | [2, 7] |

**b**

| $_{Laborer}\backslash^{Firm}$ | Gig | Employee |
|---|---|---|
| Gig | [3, 8] | [0, 0] |
| Employee | [0, 0] | [6, 2] |

**Figure 1.** Theoretical GameState pair payoff matrices used in demonstrations. (**a**) Bear market GameState; (**b**) bull market GameState.

In the bear market GameState, $a_l > d_l$ and $a_f < d_f$. The laborer receives a higher payoff for competing as a gig worker (payoff: 9 vs. 2) and the firm receives a higher payoff for hiring an employee (payoff: 3 vs. 7).

In the bull market GameState, $a_l < d_l$ and $a_f > d_f$. The laborer receives a higher payoff for competing as an employee (payoff: 3 vs. 6) and the firm receives a higher payoff for hiring a gig worker (payoff: 8 vs. 2).

#### 3.1.3. Saddle Point Geographies

In our phase diagrams, $y_1$ denotes firm strategy for gig and $x_1$ denotes laborer strategy for gig, consistent with our replicator equations.

Payoff relationships determine the geography of the saddle point. We list the general conditions for saddle point positions in regard to our quadrant legend, see Figure 2.

Quadrant  I: $a_l < d_l$ and $a_f < d_f$;
Quadrant  II: $a_l < d_l$ and $a_f > d_f$;
Quadrant  III: $a_l > d_l$ and $a_f > d_f$;
Quadrant  IV: $a_l > d_l$ and $a_f < d_f$.

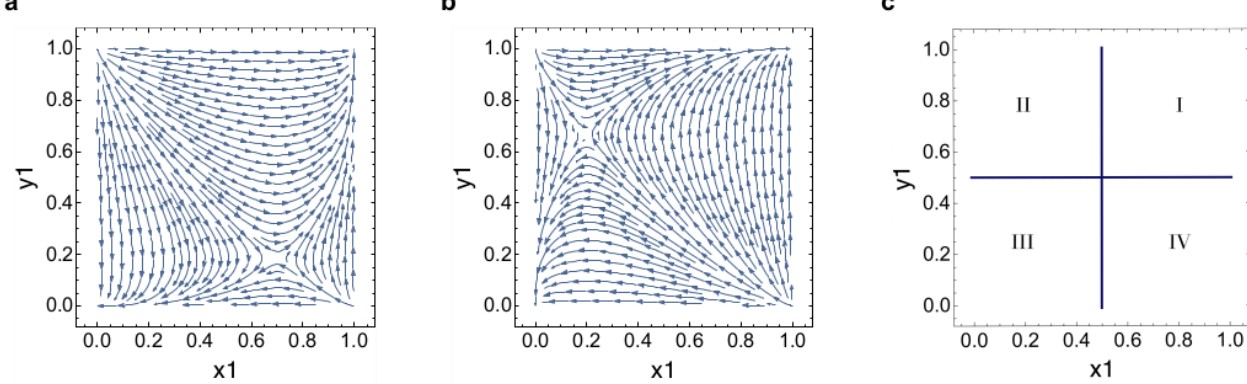

**Figure 2.** Saddle point geographies with theoretical GameState payoffs, see Figure 1. (**a**) Bear GameState, $n = 0$; (**b**) bull GameState, $n = 1$; (**c**) quadrant legend. $x_1$ and $y_1$ denote laborer and firm preference for gig work, respectively, where the value of 1.0 represents a universal gig strategy and the value of 0.0 represents a universal employee strategy.

Indeed, the saddle point for the theoretical bear GameState is located at $(\frac{7}{10}, \frac{2}{11})$ in quadrant IV. The saddle point for the theoretical bull GameState sits at $(\frac{1}{5}, \frac{2}{3})$ in quadrant II.

### 3.1.4. Attractor Arc, Driven Oscillation and Trapping Zones
Attractor Arc

In our model, we refer to our model's saddle point as an *attractor* (more strictly, which acts as an attractor for some trajectories and a repellor for others), a term we adopt and extend from the mathematical study of dynamical systems which describes a locale in the phase space that the system gravitates towards [59,60].

For a dynamical system with an environment $n$ that does not change states as a function of time, $\dot{n} = 0$, the system will evolve to one of the two stable equilibria at $(0,0)^*$ or $(1,1)^*$ dependent on initial condition; the system represents the composition of gig strategies in firm ($y$) and laborer ($x$) populations. In Figure A1 in Appendix A.1.3, we demonstrate this concept with $n = 0$, denoting the bear market GameState, and initial conditions $(\frac{1}{4}, \frac{1}{4})$ and $(\frac{3}{4}, \frac{3}{4})$ to show two evolutionary paths.

For a dynamical system with an environment $n$ that evolves as a function of time, $\dot{n} \neq 0$, phenomena of interest is centered around the *attractor arc*. The attractor arc represents the entirety of possible attractor (saddle point) positions given $n \in [0,1]$. Mathematically speaking, the attractor arc defined in the present work can be viewed as an invariant manifold; if starting from one given point at the attractor arc, the system's trajectories under changing market conditions will remain on the arc. To graphically represent the attractor arc, we superimpose our theoretical bear, $n = 0$, and bull, $n = 1$, GameState phase diagrams and plot the saddle points for all $n \in [0,1]$. The phase diagram for $n = 0$ is superimposed in orange while that of $n = 1$ is superimposed in blue. Below, the attractor arc is represented in purple. It is important to note that while this superimposed visual exhibits five reference saddle points, only one saddle point exists at any given time $t$.

We attain the preceding arc (see Figure 3) by collapsing the three dimensional $[x_1, y_1, n]$ attractor arc (see Figure 4) onto $x_1$ and $y_1$ dimensions.

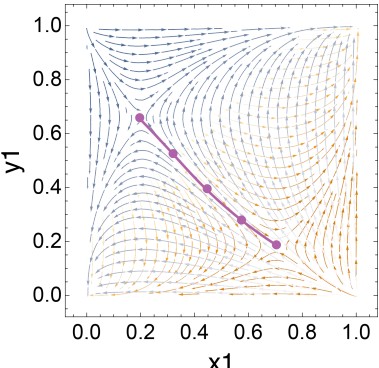

**Figure 3.** 2D attractor arc mapping on superimposed theoretical GameState payoff when $n = 0$ and $n = 1$, see Figure 1. The attractor arc represents the entirety of possible attractor positions given $n \in [0, 1]$. Reference points on the attractor arc demonstrate attractor positions when $n = 0$, $n = 0.25$, $n = 0.5$, $n = 0.75$ and $n = 1$.

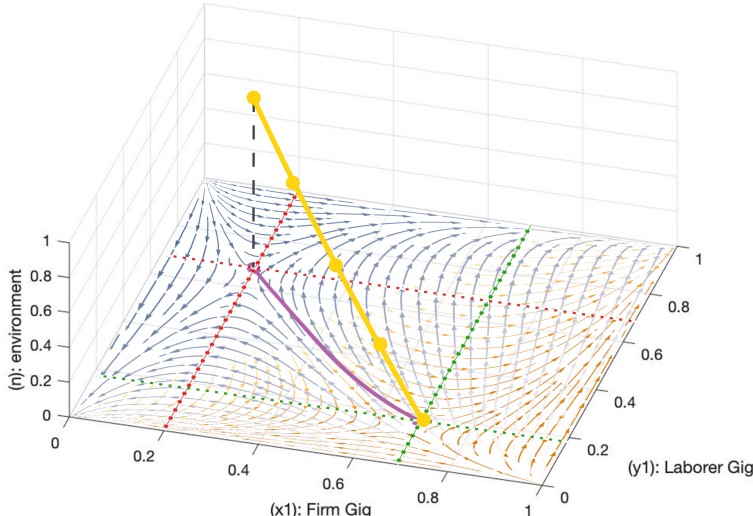

**Figure 4.** 3D attractor arc. The 3D arc is represented in yellow with reference attractor positions when $n = 0$, $n = 0.25$, $n = 0.5$, $n = 0.75$ and $n = 1$. The projected 2D arc is represented in purple, consistent with the antecedent diagram, see Figure 3. Nullclines for $n = 0$ and $n = 1$ are illustrated in green and red, respectively.

For our demonstrations, we apply a simple step-wise function for $\dot{n}$ such that the environment instantaneously alternates between $n = 0$ and $n = 1$ every 5 time units, see Figure A2 in Appendix A.1.3. Regarding the economy, this implies that a bull market will persist for 5 time units before transitioning to a bear market which will also persist for 5 time units. For clarity, we plot our selected $\dot{n}$ to help visualize the rate of change for the environment. Notably, our step-wise $\dot{n}$ implies that the attractor will jump from the two extremes of the attractor arc corresponding to $n = 0$ and $n = 1$. While we provide a reference attractor arc in all demonstrations, our $\dot{n}$ implies the attractor will not take an intermediary position on the arc. We note that the selected $\dot{n}$ is solely used in demonstrations to help illustrate each dynamic in a simple manner. More generally, our framework can be applied to general time-varying dynamics $n(t)$ of market cycles that are exogenously driven or endogenously by the collective gig strategy evolution [49,51]. In reality, $n(t)$ can take any function, which we discuss further in later sections.

### 3.1.5. Shepherding Attractors, Driven Oscillation and Trapping Zones

For a given pair of GameStates, $\dot{n}$ determines the orbit and moving speed of the attractor. As the attractor orbits the attractor arc, the attractor's oscillation can drive the system to oscillate as well. We refer to this as a driven oscillation. Near the attractor arc, there exists a trapping zone. We introduce the trapping zone as a region where the system can persist in a pseudo-stable state for numerous cycles of environment change. We highlight this kind of trapping zone (in the sense that the dynamics can oscillate within this "attraction" zone in between market conditions) using nullclines of the systems at market conditions ($n = 1$ and $n = 0$), which fully characterize the regions of pseudo-stable states admitting such oscillations arising from market condition changes (see Figures 4 and 5). Here, the attractor has a shepherding role. In order for the attractor to herd the system for numerous periods, $\omega \neq 0$ must be small enough compared to $\dot{n} \neq 0$ such that the system does not escape the ends of the attractor arc. A simple analogy can help elucidate this concept. The attractor behaves as a shepherd who can only move along one line, the attractor arc. The system behaves like a sheep that is running towards or away from the shepherd, depending on the orientation of the attractor arc. The shepherding attractor must move from one end of the arc to the other faster than the sheep in order to trap it. If the sheep reaches an escape boundary such that the shepherding attractor can not keep up, it will escape and end up at one of the two stable equilibria at $(0, 0)^*$ or $(1, 1)^*$. Escape from the trapping zone depends on the non-trivial relationship between $\dot{n}$ and $\omega$. Therefore, given $\omega$ is very small, such that the system is evolving much slower than the attractor, the trapping zone behaves as a pseudo-stable equilibrium between a pair of GameStates. Without environmental changes, the system remains stationary at the attractor arc, and it will slowly escape to one of the stable equilibria.

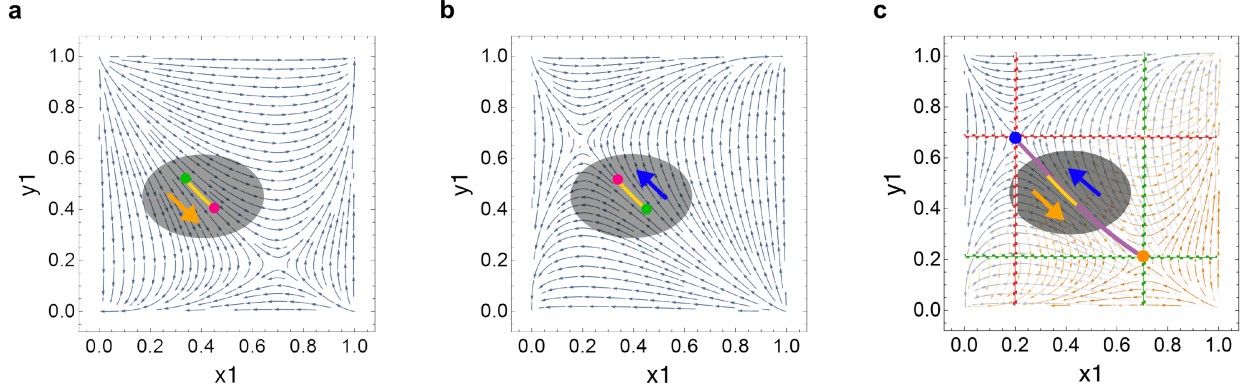

**Figure 5.** Concept visuals: Shepherding attractors and driven oscillation. (**a**) Evolution in bear market; (**b**) evolution in bull market; (**c**) driven oscillation. In (**a**,**b**), we plot the evolutionary trajectories for a bear and bull market. For each phase diagram, green denotes initial condition, red denotes ending destination and yellow denotes the evolutionary path. In (**c**), a reference attractor arc is plotted in purple and attractor positions at $n = 0$ and $n = 1$ are represented in orange and blue, respectively. The trapping zone orbit is plotted in yellow. The opaque black ellipse is a background element for visual contrast. This oscillation models $\omega = 0.5$ and initial conditions $n = 1$ and $(0.45, 0.4)$, the attractor position when $n = 0.5$. In this figure, we use a relatively large $\omega$ for the purpose of visualizing the evolution in (**a**,**b**). In (**c**), nullclines for $n = 0$ and $n = 1$ are illustrated in green and red, respectively. The central region demarcated by the nullclines is an attraction zone where trapping behavior is possible.

#### Escape and Implications

Assuming that the system has previously existed by oscillating in the trapping zone, escape is possible if there is a perturbation that changes $\dot{n}$ and or $\omega$ such that the system reaches escape boundary. Once the system reaches escape boundary, the system will eventually escape the trapping zone to one of the stable equilibria at $(0, 0)^*$ or $(1, 1)^*$, see Appendix A.2.

With escape, it is important to note that initial condition is crucial in determining which stable equilibrium the system escapes to. If $\omega$ increases twenty-fold such that the system reaches escape boundary at the start of a bear market, $n = 0$, rather than at the start of a bull market, $n = 1$, the system evolves to $(0, 0)^*$ rather than $(1, 1)^*$. A claim based on which of the two stable equilibria the system escapes to is indefensible, as this result is subject to the initial conditions. As such, we theorize the possibility of escape but do not run our models to make a claim for a specific escape destination. Therefore, we can only conclude that changes in $\dot{n}$ and $\omega$ can allow the system to reach escape boundary and result in an accelerated escape to one of the two stable equilibria. However, we can not conjecture which stable equilibrium the system escapes to.

Selection of Initial Conditions

When applying this model, it is unfitting to prepare any arbitrary initial condition because different initial conditions can result in different evolutionary outcomes, see Appendix A.2.3. Therefore, all findings or claims fixating on a specific ESS can be countered with the selection or preparation of another initial condition.

In our model, we presume that some co-existence of gig and employee strategies has always been present in the labor market. Our evolutionary system informs us that if the labor market consisted of only one type of worker (gig or employee) in the past, there would be no co-existence of gig and employee strategies today, as the system would have remained fixated on that ESS; therefore, we reason that the present day co-existence of gig and employee strategies necessitates a historical co-existence of gig and employee strategies.

Mathematically, this implies that our system has always been "trapped" in a state of oscillatory dynamics up until the observable present-day. Appropriately, in this work, we have defined the mechanism that "traps" the system in this pseudo-stable state of gig and employee co-existence.

Regarding the gig economy, we assume that observable fluctuations in labor strategies reflect the system oscillating within the trapping zone (i.e., what we observe is pseudo stable state at all times). It is sensible for our system to evolve within the pseudo-stable trapping zone as this represents the present-day domain of oscillatory dynamics and observable co-existence of gig and employee strategies. Therefore, any point in the trapping zone is a suitable initial condition. In our models, we use the attractor position at $n = 0.5$, the midway point between a bear and bull market transition, as an estimator for a point in the trapping zone.

Attractor Arc Drift and Tilt

If we consider payoffs as a function of time, $\dot{A}, \dot{B} \neq 0$, the attractor arc itself will evolve, see Figure 6. Accordingly, this implies the trapping zone will change position with the attractor arc because the system's orbit is a driven oscillation. Assuming the system exists by always oscillating in the pseudo-stable trapping zone, evolving payoffs can help explain how the *system's orbit*, an orbit in the trapping zone, can move around the phase space. The shape and orientation of the arc at any given time $t$ depends on $\dot{A}$ and $\dot{B}$. In later sections, we investigate payoff operations that cause the attractor arc to drift (change position in the phase space) and tilt (change orientation in the phase space).

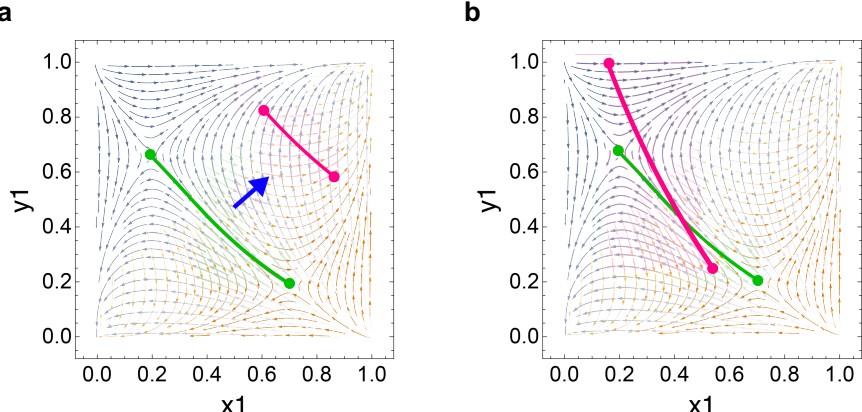

**Figure 6.** Concept visuals: Attractor arc drift and arc tilt. (**a**) Attractor arc drift $\dot{A}, \dot{B} \neq 0$; (**b**) attractor arc tilt $\dot{A}, \dot{B} \neq 0$. In (**a**), the green arc applies the theoretical GameState pair payoff, see Figure 1, and the red arc applies a high employee payoff matrix operation, see Figure 12. In (**b**), the green arc applies the theoretical GameState pair payoff, see Figure 1, and the red arc applies a lenient policy matrix operation, see Figure 14.

### 3.2. Market Influences on Firm and Laborer Gig Preference

In this section, we introduce a generalized framework that demonstrates how evolving market conditions effectuate oscillating labor dynamics. First, we demonstrate how oscillating dynamics can be analyzed in the context of the gig economy by presenting two theoretical examples and interpreting their contrasting dynamics. We note that the theoretical payoffs used in demonstrations have no relationship to any specific labor economy. After demonstrating how dynamics can be interpreted, we establish a generalized framework for market influenced oscillatory dynamics that can be applied to general payoff values. Finally, we discuss payoff generation by exploring labor considerations that influence firm and laborer payoffs.

#### 3.2.1. Interpretations of Market Influenced Dynamics

To demonstrate how dynamics can be interpreted, we present two theoretical examples where observed labor strategy densities evolve over the period of three market cycles. The two examples leverage theoretical payoff matrices, specifically chosen to illustrate contrasting dynamics.

#### Market Influence on Labor Dynamics, Example No. 1

In our first example, see Figure 7, we model market influenced oscillatory dynamics over three market cycles with the theoretical payoff matrices from Figure 1. The simulation begins at the start of a bull market, $n = 1$. It is important to note that the geographical relationship between attractor positions at $n = 0$ and $n = 1$ determines the direction of dynamics. In this specific example, the attractor arc is negatively sloped with the attractor position at $n = 0$ and $n = 1$ located in quadrant IV and quadrant II, respectively. Attractor positions at $n = 0$ and $n = 1$ are represented in orange and blue, respectively. Directional dynamics during bear and bull markets are represented by the orange and blue arrows, respectively.

On one hand, during a bull market, firm preference for gig strategies increases while laborer preference for gig strategies decreases. On the other hand, during a bear market, firm preference for gig strategies decreases while laborer preference for gig strategies increases. Here, we observe mismatching oscillatory dynamics for firm and laborer gig preferences when market conditions change.

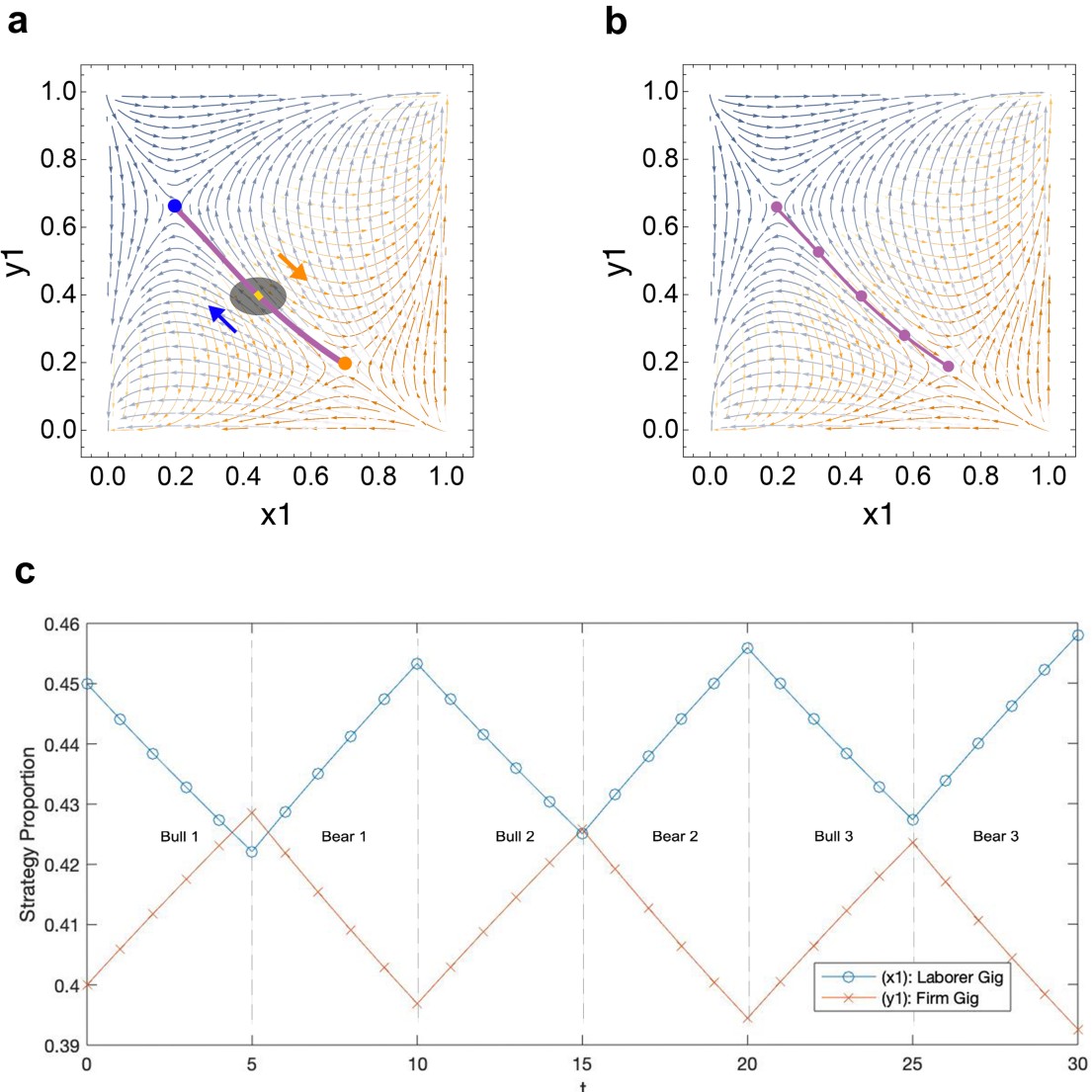

**Figure 7.** Theoretical market demonstration No. 1: Evolution of strategy densities under market influence with initial conditions $(0.45, 0.40)$, the attractor position at $n = 0.5$, an approximation for a point in the trapping zone; $n = 1$, a bull market; $\omega = 0.01$; and theoretical payoff matrices from Figure 1. (**a**) Trapping zone orbit. (**b**) Attractor arc. (**c**) Labor strategy oscillation over three market periods. In (**a**), the trapping zone orbit is plotted in yellow, and attractor positions at $n = 0$ and $n = 1$ are represented in orange and blue, respectively. Directional dynamics during bear and bull markets are represented by the orange and blue arrows, respectively. In (**b**), we plot a reference attractor arc in purple with attractor positions when $n = 0$, $n = 0.25$, $n = 0.5$, $n = 0.75$ and $n = 1$. (**c**) visualizes the fluctuation in firm and laborer preferences for gig strategies over three market cycles.

Market Influence on Labor Dynamics, Example No. 2

In our second example, see Figure 8, we model market influenced oscillatory dynamics over three market cycles with the theoretical payoff matrices from Figure A6 in Appendix B. The simulation begins at the start of a bull market, $n = 1$. In this example, the attractor arc is positively sloped with the attractor position at $n = 0$ and $n = 1$ located in quadrant III and quadrant I, respectively.

In a bull market, both firm and laborer preferences for gig strategies increase. During a bear market, both firm and laborer preferences for gig strategies decrease. Here, we observe matching oscillatory dynamics for firm and laborer gig preferences when market conditions change.

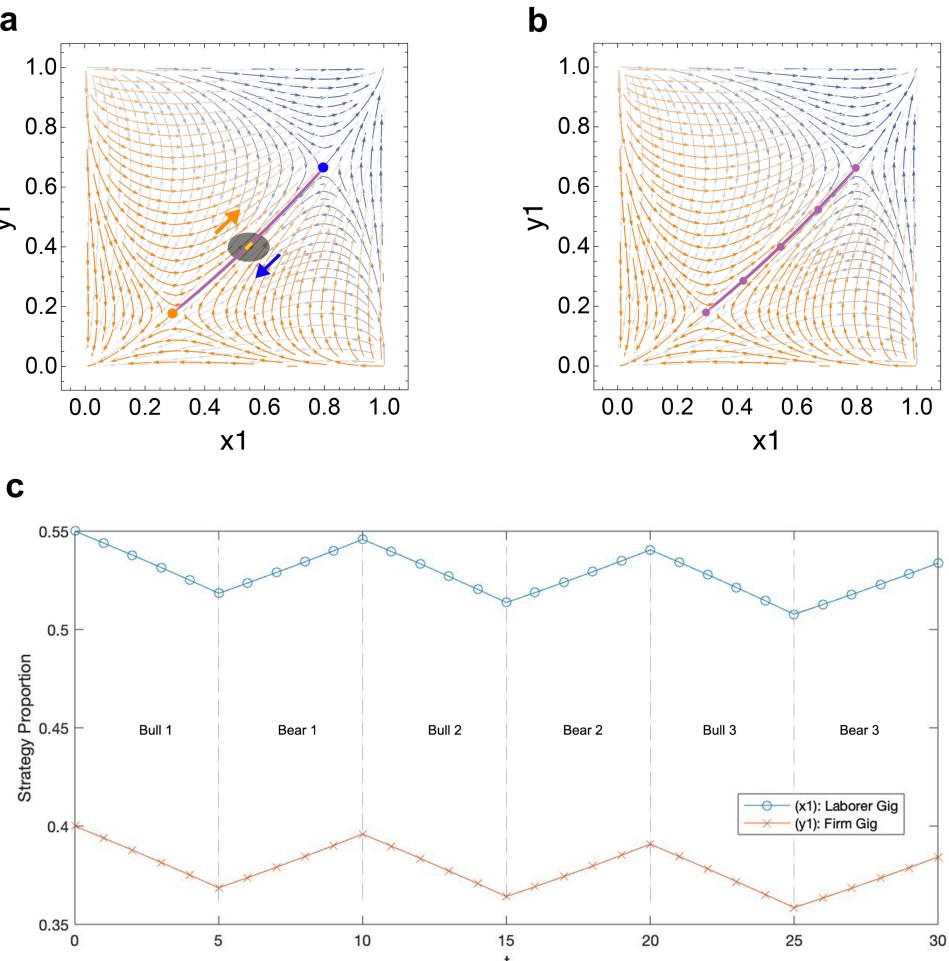

**Figure 8.** Theoretical market demonstration No. 2: Evolution of strategy densities under market influence with initial conditions $(0.55, 0.40)$, the attractor position at $n = 0.5$, an approximation for a point in the trapping zone; $n = 1$, a bull market; $\omega = 0.01$; and theoretical payoff matrices from Figure A6 in Appendix B. (**a**) Trapping zone orbit. (**b**) Attractor arc. (**c**) Labor strategy oscillation over three market periods. In (**a**), the trapping zone orbit is plotted in yellow, and attractor positions at $n = 0$ and $n = 1$ are represented in orange and blue, respectively. Directional dynamics during bear and bull markets are represented by the orange and blue arrows, respectively. In (**b**), we plot a reference attractor arc in purple with attractor positions when $n = 0$, $n = 0.25$, $n = 0.5$, $n = 0.75$ and $n = 1$. (**c**) visualizes the fluctuation in firm and laborer preferences for gig strategies over three market cycles.

### 3.2.2. Generalized Framework for Market Influenced Oscillatory Dynamics

In our two examples, we illustrate contrasting market influenced oscillatory dynamics and provide basic interpretations. Indeed, dynamics are determined based on the geographical relationship between the attractor location at $n = 0$ and $n = 1$. Revisiting Section 3.1.3, we note that attractor positions at $n = 0$ and $n = 1$, which constitute the arc, are governed by payoff inequalities.

The proposed evolutionary model can be used to study any labor economy, whether it be that of a country, a skill-partitioned subset of a country or a single firm. For each labor economy, however defined, firms and laborers may have unique payoff considerations, resulting in labor economy specific payoff inequalities. Economy-specific payoffs govern the attractor arc and influence the observed dynamics of the studied labor ecosystem.

In this section, we present a generalized framework for market influenced oscillatory dynamics that is indiscriminate of labor economy specific payoffs. To begin, we rephrase

bull market payoffs in terms of bear market payoffs and a market $\delta$, which captures how labor preference varies from a bear to bull market (see Figure 9).

**a**

| $_{Laborer}\backslash^{Firm}$ | Gig | Employee |
|---|---|---|
| Gig | $[a_{l0}, a_{f0}]$ | $[0, 0]$ |
| Employee | $[0, 0]$ | $[d_{l0}, d_{f0}]$ |

**b**

| $_{Laborer}\backslash^{Firm}$ | Gig | Employee |
|---|---|---|
| Gig | $[a_{l0} + \delta a_l^n,\ a_{f0} + \delta a_f^n]$ | $[0, 0]$ |
| Employee | $[0, 0]$ | $[d_{l0} + \delta d_l^n,\ d_{f0} + \delta d_f^n]$ |

**Figure 9.** $n = 1$ payoffs represented in terms of $n = 0$ payoffs. (**a**) Bear market, $n = 0$. (**b**) Bull market, $n = 1$. We rephrase bull market payoffs in terms of bear market payoffs and a market $\delta$, which captures how labor preference varies from a bear to bull market. For simplification purposes, we assign all mismatching strategies a payoff of 0 as no mutual labor agreement is made between firm and laborer.

Next, we normalize payoffs during a bear market, $n = 0$.
$$a_{l0} = a_{f0} = d_{l0} = d_{f0} = 1$$

Once normalized, the attractor location when $n = 0$ is located at $(\frac{1}{2}, \frac{1}{2})$ in the phase space. The payoff inequalities between market $\delta$ terms for each firm and laborer strategy determine the attractor location at $n = 1$. Therefore, the relationship between market $\delta$ terms illustrate how the attractor position at $n = 1$ is geographically positioned relative to that at $n = 0$.

In Figure 10, we present a generalized map of directional dynamics through the lens of a normalized $n = 0$ payoff.

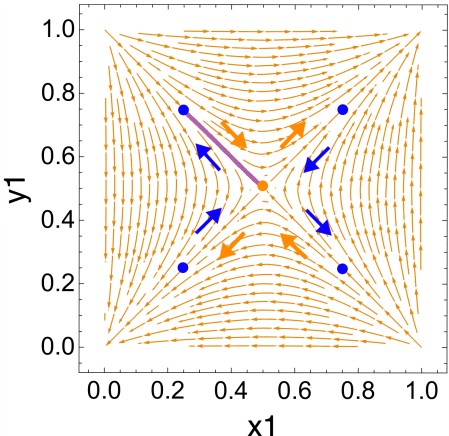

**Figure 10.** Directional framework for market influenced dynamics with normalized bear market payoffs. Directional dynamics are presented through the lens of a normalized $n = 0$ payoff. The attractor position at $n = 0$ is represented in orange. Four theoretical attractor positions at $n = 1$, represented in blue, are depicted in quadrants I through IV. Directional dynamics during bear and bull markets are represented by the orange and blue arrows, respectively. A single attractor arc is illustrated in purple to demonstrate when the attractor position at $n = 1$ is located in quadrant II in respect to a normalized attractor position at $n = 0$. Here, directional dynamics are similar to our first theoretical example, see Figure 7.

### 3.2.3. Payoff Generation

To generate strategy payoffs for a specific labor economy, a combination of firm-specific and laborer-specific considerations must be evaluated. For example, firm payoffs may reflect the cost of labor, worker reliability, enterprise-scalability, talent acquisition costs and marginal benefits accrued from labor flexibility. For the laborer, compensation, career mobility, stress, benefits, status and flexibility are some examples of inputs that can influence strategy payoffs.

Weights of the selected labor considerations may change in bear and bull markets to reflect how firms and laborers adapt to evolving economic environments.

In the Supplementary Materials, we provide four examples of payoff generation in which we scope the labor economy to a small low-skill firm, a small high-skill firm, a large low-skill firm and a large high-skill firm. We derive payoffs from a confluence of labor considerations across market conditions and provide a comparative analysis for the simulated labor economies.

*3.3. Technology Influences on Firm and Laborer Gig Preference*

After we demonstrate that the system oscillating within the trapping zone reflects observable fluctuations in gig strategy densities across market conditions, we proceed to explore the role of technology in the gig economy.

In this theoretical extension, we introduce a framework that demonstrates how technology influences labor payoffs and the growth of the gig economy. To begin, we analyze the nature in which evolving payoffs, $\dot{A}, \dot{B} \neq 0$, shift the position of the attractor arc. We use the theoretical GameState pair, see Figure 1, as our reference payoff matrix pair. Let us assume that the reference payoff matrix pair represents present-day payoffs. As shown below, the reference attractor arc is rendered in yellow.

To demonstrate the position of the attractor arc when gig strategies offer high payoffs, we add $\delta a^q$ to the reference payoff matrix pair for all matching gig strategies where $\delta a^q$ denotes a technology-driven increase in gig strategy payoffs, see Figure 11. The attractor arc for high gig payoffs is represented in blue, see Figure 13a.

**a**

| $_{Laborer}\backslash^{Firm}$ | Gig | Employee |
|---|---|---|
| Gig | $[a_{l0} + \delta a^q, \, a_{f0} + \delta a^q]$ | $[b_{l0}, b_{f0}]$ |
| Employee | $[c_{l0}, c_{f0}]$ | $[d_{l0}, d_{f0}]$ |

**b**

| $_{Laborer}\backslash^{Firm}$ | Gig | Employee |
|---|---|---|
| Gig | $[a_{l1} + \delta a^q, \, a_{f1} + \delta a^q]$ | $[b_{l1}, b_{f1}]$ |
| Employee | $[c_{l1}, c_{f1}]$ | $[d_{l1}, d_{f1}]$ |

**Figure 11.** High gig payoff, matrix operation. (**a**) High gig payoff, $n = 0$; (**b**) high gig payoff, $n = 1$.

To demonstrate the position of the attractor arc when employee strategies offer high payoffs, we add $\delta d^q$ to the reference payoff matrix pair for all matching employee strategies where $\delta d^q$ denotes a technology-driven increase in employee strategy payoffs, see Figure 12. The attractor arc for high employee payoffs is illustrated in red, see Figure 13b.

**a**

| $_{Laborer}\backslash^{Firm}$ | Gig | Employee |
|---|---|---|
| Gig | $[a_{l0}, a_{f0}]$ | $[b_{l0}, b_{f0}]$ |
| Employee | $[c_{l0}, c_{f0}]$ | $[d_{l0} + \delta d^q, \, d_{f0} + \delta d^q]$ |

**b**

| $_{Laborer}\backslash^{Firm}$ | Gig | Employee |
|---|---|---|
| Gig | $[a_{l1}, a_{f1}]$ | $[b_{l1}, b_{f1}]$ |
| Employee | $[c_{l1}, c_{f1}]$ | $[d_{l1} + \delta d^q, \, d_{f1} + \delta d^q]$ |

**Figure 12.** High employee payoff, matrix operation. (**a**) High employee payoff, $n = 0$; (**b**) high employee payoff, $n = 1$.

3.3.1. Technology and the Neoteric Growth of the Gig Economy

In recent decades, the gig economy has ballooned from relative obscurity to more than one third of the labor market [3]. Mapped to our model, this growth implies that the attractor arc evolved from a region near $(0,0)^*$ towards $(1,1)^*$; this is pictured as a shift from the blue arc to the yellow arc, indicating an increase in gig workers.

Our model suggests that the premature gig economy (see Figure 13a), which consisted of fewer workers pursuing non-standard or gig employment, involved gig labor that averaged higher payoffs compared to employee labor (see Figure 11). Since payoff is determined by compensation, high payoff implies high compensation. Appropriately, when gig payoffs are very high, each company can afford to hire a small amount of these

elite, skilled non-standard workers. This explains why the attractor arc for high gig payoffs is located near $(0,0)^*$, indicating a labor composition consisting of few gig workers. Often, firms recruit highly skilled non-standard workers to provide rare, distinctive skills and abilities that are unavailable within the company [61]. Previous studies suggest that highly skilled workers benefit more from the flexibility provided by non-standard employment and possess increased negotiating leverage due to their desirable skill-sets [61,62]. Highly skilled workers may be more inclined to pursue non-standard employment due to this increased flexibility [63]. Contrarily, low-skill workers pursuing non-standard employment are more exposed to the precarity of evolving market conditions and possess less bargaining power [64,65].

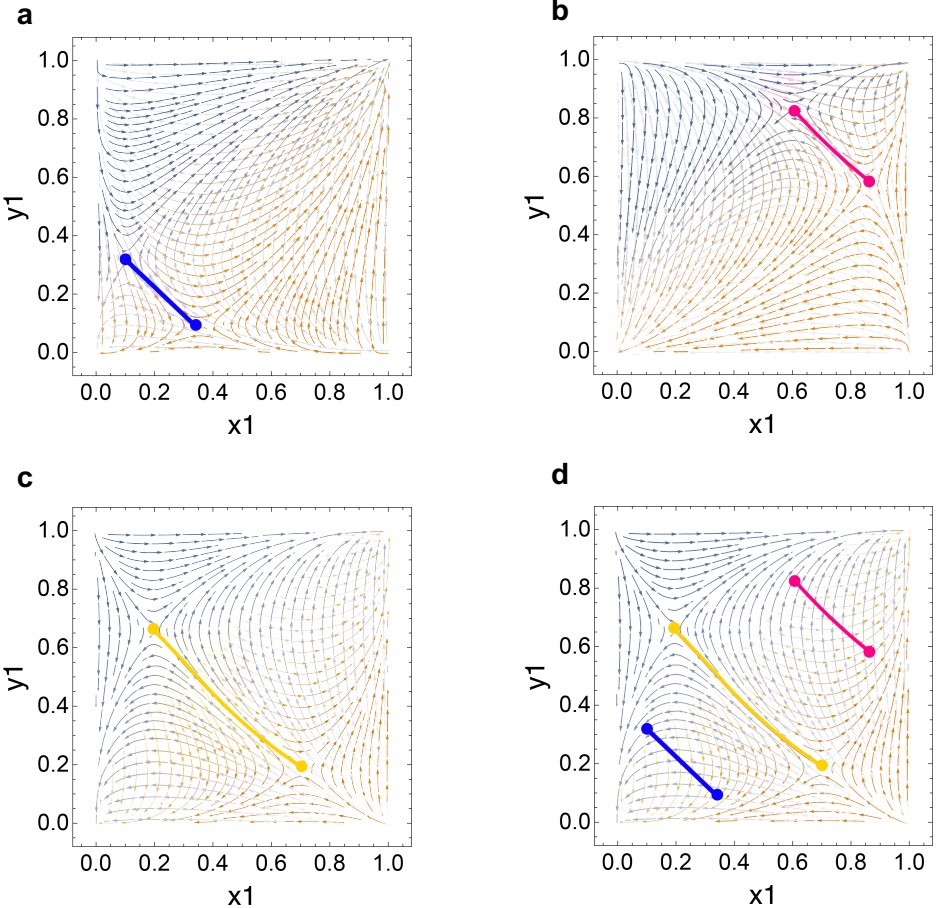

**Figure 13.** Attractor arc drift transformations. (**a**) Arc transformation with high gig payoff matrix operation, see Figure 11; theoretical GameState pair, see Figure 1; and $\delta a^q = \delta d^q = 10$. (**b**) Arc transformation with high employee payoff matrix operation, see Figure 12; theoretical GameState pair, see Figure 1; and $\delta a^q = \delta d^q = 10$. (**c**) Reference attractor arc with theoretical GameState pair, see Figure 1. (**d**) Composite diagram with arcs (**a**–**c**).

More recently, technology has enabled firms to outsource more piecemeal work to low-skill workers. Existing literature suggests that technology reduces task complexity, enabling workers with fewer skills to complete these jobs [66,67]. Without contemporary gig platforms, there was a higher barrier to discover and secure auxiliary work arrangements. Digital platforms enable low-skill workers to feasibly participate in the gig economy by coordinating and matching streams of labor supply and demand and providing workers with a sustained cascade of work opportunity. Examples of such technologies include ride-sharing and last-mile delivery apps such as Uber, Lyft and DoorDash as well as freelancing websites such as Upwork, all of which introduce mostly low-skill, low-payoff

workers to the gig economy. While some readers may regard sharing-economy applications as platform operations, drivers as gig workers and customers as the firm who employs the driver, we clarify our interpretation of the sharing-economy platform as the firm. As low-skill gig workers such as Uber drivers flooded the gig economy, gig payoffs decreased relative to employee payoffs. This is consistent with our model as the attractor arc shifts towards $(1, 1)^*$ from the blue to yellow arc during this development, reflecting the neoteric growth of the modern gig economy.

### 3.3.2. Technological Implications on the Future of the Gig Economy

Notional future growth of the gig economy is represented by the evolution from the yellow arc to the red arc. Per our model, as employee payoffs increase relative to gig payoffs, the attractor arc nears $(1, 1)^*$; this implies that the labor market consists of mostly gig workers and few employees. Some ride-sharing firms may already example such distinct gig-employee bifurcation consistent with an arc positioned near $(1, 1)^*$. For instance, Uber's personnel consists of many low payoff gig drivers, and relatively few high payoff engineers, managers and executives. Such a distribution is reflected in our model as we observe an attractor arc position higher up on the $y_1$ axis for low-skill firms, implying a workforce with a higher density of gig laborers.

There are cogent reasons to believe that the gig economy might either decrease or increase in size, a tension we aim to inform. We offer model-informed explanations that acknowledge the two competing logics. In order for the gig economy to continue growing, employee payoffs must increase relative to gig payoffs. Such a development implies that high skill work must benefit gig roles such that executives, the highest paid individuals, are the only employees remaining in an enterprise. In the current enterprise structure, there are numerous obstacles facing such a workforce transformation. While low-skill firms compete on pricing, high skill firms compete on talent. Thus far, most gig-dominant firms are low-skill firms such as Uber and Lyft which leverage commodity skill workers to operate their services. On the other hand, the notion of ubiquitous high-skill gig work faces the legal and strategic complication of trade secrets, non-disclosure agreements, non-competes and other intellectual property complexities. Further, there is growing consensus that artificially intelligent machines will replace many processes currently fulfilled by commodity-skill human operators. Resultantly, low-skill gig workers such as Uber drivers will be displaced as a part of this technological transformation, signaling a future contraction in the present day low-skill dominant gig economy. The question is whether these displaced workers will find new roles as employees or gig workers.

There are also compelling reasons to believe that the gig economy will continue growing. Researchers have conjectured that workers displaced by AI technologies will find roles in which they supervise machines and fulfill other more creative responsibilities [68]. Creative roles are a suitable fit for the gig economy as these positions champion worker flexibility. While ride-share companies like Uber and Lyft may decrease their gig application, the freelancing cohort of the gig economy may potentially continue growing. Laying the foundation for increasingly dynamic careers, education and learning opportunities augmented by technology may also play a role in re-skilling or up-skilling the future workforce. Further, the future may entail a re-constitution of enterprise with pioneering frontier technologies, decentralization and remote work arrangements. A reconstitution of policy structures can also play a role in the regulation and protection of trade secrets, all of which may support adoption of ubiquitous high-skill gig work.

The work and enterprise structures of the future depend on a dizzying constellation of cultural and technological developments, rendering it difficult to speculate the future direction of the gig economy. While we address the competing logics, we do not state a specific preference for future gig economy growth or contraction. We hope that our model extension can inform the discussion by providing a new payoff framework that can be applied when thinking about technology's role in the growth of the gig economy.

### 3.4. Policy Influences on Firm and Laborer Gig Preference

Using a similar approach, we also explore policy influences on labor strategies by applying an evolving-payoff framework. While the gig economy has been viewed as beneficially transformative to some, others share a more precarious disposition regarding its economic imbalances. For researchers, policy makers and industrialists alike, there exists a tension as to whether or how to regulate the gig economy. In the context of the labor market, policy behaves as a mechanism that can transfer risk and economic burdens between firms and laborers [25,28,29].

During periods of lenient policy regulation, firms can take advantage of regulatory ambiguity and exploit gig workers. On the other hand, gig laborers are unprotected and must tolerate firm expectations. To model the payoff during a period of lenient policy ordinance, we subtract $\delta a^p$ from the laborer's gig payoff and add $\delta a^p$ to the firm's gig payoff where $\delta a^p$ denotes the payoff from transferred economic risk, see Figure 14. The attractor arc for lenient policy ordinance is represented in red, see Figure 16a.

**a**

| $Laborer\backslash^{Firm}$ | Gig | Employee |
|---|---|---|
| Gig | $[a_{l0} - \delta a^p,\, a_{f0} + \delta a^p]$ | $[b_{l0}, b_{f0}]$ |
| Employee | $[c_{l0}, c_{f0}]$ | $[d_{l0}, d_{f0}]$ |

**b**

| $Laborer\backslash^{Firm}$ | Gig | Employee |
|---|---|---|
| Gig | $[a_{l1} - \delta a^p,\, a_{f1} + \delta a^p]$ | $[b_{l1}, b_{f1}]$ |
| Employee | $[c_{l1}, c_{f1}]$ | $[d_{l1}, d_{f1}]$ |

**Figure 14.** Lenient policy, matrix operation. (**a**) Lenient ordinance, $n = 0$. (**b**) Lenient ordinance, $n = 1$.

In the course of strict policy enactment, governments demand firms to more closely classify gig workers as employees. For instance, a government may mandate that firms provide benefits and additional protections to gig laborers. Accordingly, gig workers benefit as they receive additional worker protections and increased welfare. To model the payoff during a period of strict policy ordinance, we add $\delta a^p$ from the laborer's gig payoff and subtract $\delta a^p$ to the firm's gig payoff, see Figure 15. The attractor arc for strict policy ordinance is represented in blue, see Figure 16b.

**a**

| $Laborer\backslash^{Firm}$ | Gig | Employee |
|---|---|---|
| Gig | $[a_{l0} + \delta a^p,\, a_{f0} - \delta a^p]$ | $[b_{l0}, b_{f0}]$ |
| Employee | $[c_{l0}, c_{f0}]$ | $[d_{l0}, d_{f0}]$ |

**b**

| $Laborer\backslash^{Firm}$ | Gig | Employee |
|---|---|---|
| Gig | $[a_{l1} + \delta a^p,\, a_{f1} - \delta a^p]$ | $[b_{l1}, b_{f1}]$ |
| Employee | $[c_{l1}, c_{f1}]$ | $[d_{l1}, d_{f1}]$ |

**Figure 15.** Strict policy, matrix operation. (**a**) Strict ordinance, $n = 0$. (**b**) Strict ordinance, $n = 1$.

The Impact of Regulation on Labor Strategy Sensitivities

While the position of the attractor arc is a suitable proxy for the position of the trapping zone, arc orientation does not always represent the orientation of the trapping zone. As the slope of the arc increases and the arc becomes more vertical, the slope of the trapping zone decreases and becomes more horizontal. Conversely, as the slope of the arc decreases and the arc becomes more horizontal, the slope of the trapping zone increases and becomes more vertical. This concept is visually represented in Figure 17. On one hand, when the attractor arc becomes more vertical, laborers experience an increased sensitivity between employee and gig strategies across market cycles while firms experience a decreased sensitivity; this can be understood as oscillators in the trapping zone become elongated on the $x_1$ axis and shortened on the $y_1$ axis. On the other hand, when the attractor arc becomes more horizontal, laborers experience a decreased sensitivity between employee and gig strategies across market cycles while firms experience an increased sensitivity; this can be understood as oscillators in the trapping zone become shortened on the $x_1$ axis and elongated on the $y_1$ axis. We define an oscillator as an evolutionary orbit for the system

across market cycles. We define sensitivity as the distinction between and preference for gig or employee strategies across market conditions.

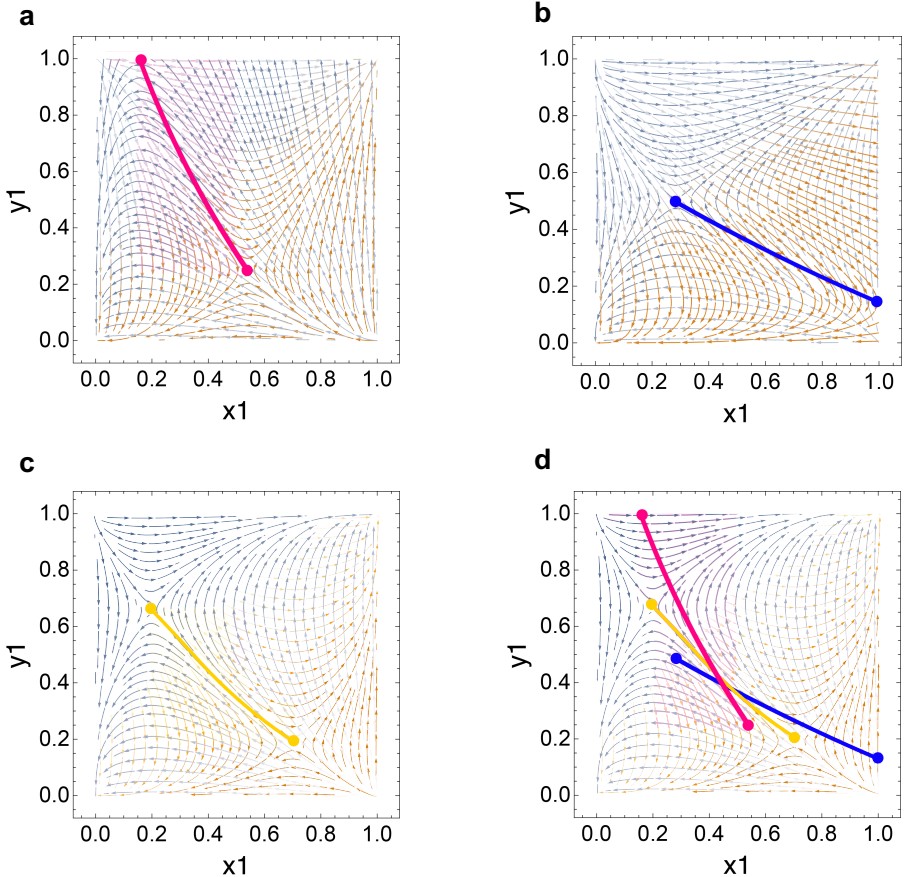

**Figure 16.** Attractor arc drift transformations. (**a**) Arc transformation with lenient policy matrix operation, see Figure 14; theoretical GameState payoff pair, see Figure 1; and $\delta a^p = 3$. (**b**) Arc transformation with strict policy matrix operation, see Figure 15; theoretical GameState payoff pair, see Figure 1; and $\delta a^p = 3$. (**c**) Reference attractor arc with theoretical GameState payoff pair, see Figure 1. (**d**) Composite diagram with arcs (**a**–**c**).

An interval of lenient policy will drive the attractor arc to increase in slope and become more vertical while strict policy will drive the arc to decrease in slope and become more horizontal. For our demonstrations, we will use our vertical attractor arc as an extreme example of lenient policy and our horizontal attractor arc as an extreme example of strict policy.

During a period of strict regulatory ordinance, the firm must pay the gig worker increased compensation, even though the gig worker provides the same quality of work as before. Therefore, the firm experiences an increased sensitivity and larger distinction between gig workers and employees. If we consider our horizontal attractor arc to be an extreme example of strict policy, we see that the $y_1$ trapping zone span is elongated while the $x_1$ trapping zone span is shortened. The longer $y_1$ trapping zone span exhibits the firm's increased sensitivity to worker type and a greater distinction between hiring gig workers or employees. For laborers, the shorter $x_1$ span signifies a decreased sensitivity for participating as a gig worker or an employee; this is a logical transformation, as strict policy mandates greater equality in the treatment of gig workers and employees, forming a strengthened gig-employee resemblance.

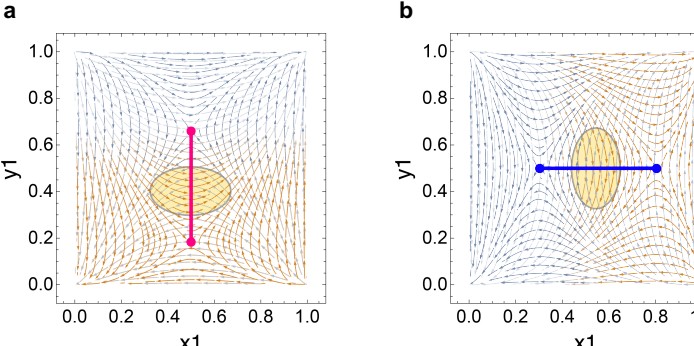

**Figure 17.** Vertical and horizontal attractor arc and trapping zone slopes. (**a**) Attractor arc using theoretical payoff pair, see Figure A7 in Appendix B. When the attractor arc is oriented vertically, the slope of the trapping zone becomes horizontal and perpendicular to the arc. (**b**) Attractor arc using theoretical payoff pair, see Figure A8 in Appendix B. When the attractor arc is oriented horizontally, the slope of the trapping zone becomes vertical and perpendicular to the arc. The opaque yellow ellipse is a background element to indicate the general region of the trapping zone. The evolutionary trajectories in both (**a**,**b**) trapping zones are orthogonal to their respective arcs.

Conversely, in a period of lenient policy denoted by the red arc, we find that the $x_1$ trapping zone span is elongated while the $y_1$ trapping zone span is shortened. Considering the lack of gig worker protections during intervals of lenient policy, it is sensible that gig workers experience increased sensitivity between worker categories without regulation, as there is greater distinction between working as an employee or a gig worker. On the other hand, firms experience a decreased sensitivity for worker type as they can take advantage of regulatory ambiguity to maximize operational efficiency.

In this theoretical extension, we assume that policy behaves as a mechanism that can shift economic burdens, represented through payoffs, between firms and laborers. We propose an evolving-payoff framework to model the impact of policy regulations on firm and worker labor strategies. Our findings inform existing literature and scholarship by demonstrating how policy transfers payoff utility and alters firm and laborer sensitivities for different labor strategies.

### 3.5. A Treble of Evolutionary Dynamics under Technology, Policy and Market Influence

In the present work, we introduce three distinct dynamics: market-driven oscillation, technology-driven arc drift and policy-driven arc tilt. Each dynamic is set in motion due to an environment-driven adjustment in payoffs. Although we introduce each dynamic individually, in reality, technology, policy and market environments evolve in concert, see Figure 18. In this section, we establish an aggregate theory that addresses the treble of evolutionary dynamics under technology, policy and market influence and discuss implications for the labor economy.

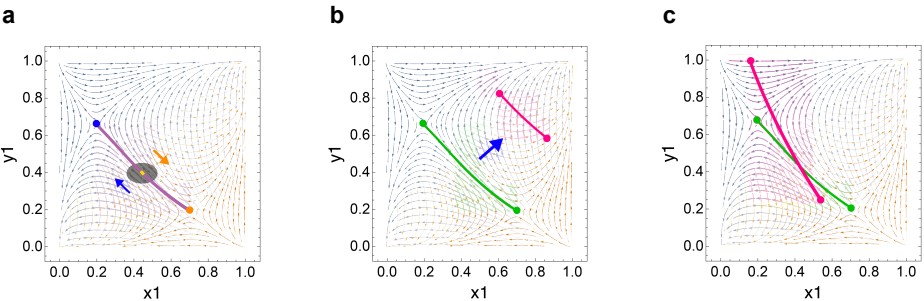

**Figure 18.** Treble of dynamics. (**a**) Market driven oscillatory dynamics. (**b**) Technology driven arc drift. (**c**) Policy driven arc tilt.

### 3.5.1. An Evolving Orbit of Forced Dynamics

To concurrently model dynamics influenced by technology, policy and markets, we introduce two new environment variables, $p$ representing the policy landscape and $q$ representing the technology landscape. Our three environment variables $n$, $p$ and $q$ all exist as functions of time. Accordingly, we reconstitute our firm and laborer payoffs as a function of $n$, $p$ and $q$ and incorporate $\delta^n$, $\delta^p$ and $\delta^q$ terms as specified in the market, technology and policy sections.

$$\Delta a_l^{(n,p,q)} = n\delta a_l^n + q\delta a^q + p\delta a^p \tag{13}$$

$$\Delta a_f^{(n,p,q)} = n\delta a_f^n + q\delta a^q + p\delta a^p \tag{14}$$

$$\Delta d_l^{(n,p,q)} = n\delta d_l^n + p\delta d^p + q\delta d^q \tag{15}$$

$$\Delta d_f^{(n,p,q)} = n\delta d_f^n + p\delta d^p + q\delta d^q \tag{16}$$

Here, $\Delta^{(n,p,q)}$ terms capture the change in strategy payoffs due to exogenous environment variables as specified in Figures 9, 11, 12, 14 and 15. $n \in [0, 1]$ denotes the market condition where 0 and 1 represent bear and bull markets, respectively. $p \in [-1, 1]$ represents the policy landscape where $-1$ and 1 represent lenient and strict policies, respectively. $q \in [0, 1]$ denotes technology adoption where 0 and 1 represent a technology landscape that favors gig and employee strategies, respectively. In respect to $n$, $p$ is a fast moving variable and $q$ is a slow moving variable to reflect the comparative pace of technology, policy and market change.

$$A(n, p, q) = \begin{pmatrix} a_{l0} + \Delta a_l^{(n,p,q)} & 0 \\ 0 & d_{l0} + \Delta d_l^{(n,p,q)} \end{pmatrix} \tag{17}$$

$$B(n, p, q) = \begin{pmatrix} a_{f0} + \Delta a_f^{(n,p,q)} & 0 \\ 0 & d_{f0} + \Delta d_f^{(n,p,q)} \end{pmatrix}^T \tag{18}$$

Here, laborer and firm payoffs are in terms of $\Delta^{(n,p,q)}$, see the equations above, where $A(n, p, q)$ and $B(n, p, q)$ denote laborer and firm payoffs, respectively. For simplification purposes, we assign all mismatching strategies a payoff of 0 as no mutual labor agreement is made between firm and laborer.

In our model, the replicator equation implies that if a specific labor strategy perishes at an ESS, it can not be revived and reintroduced to the labor economy. Therefore, the present day co-existence of gig and employee strategies informs us that the labor economy exists in a pseudo-stable state rather than at a stable equilibrium. In the present work, we present three mechanisms that work together to "trap" the system in this pseudo-stable equilibrium which manifests as an evolving orbit of forced dynamics.

We note that market influenced driven-oscillation as a lone dynamic is unstable over a sustained period of time. In Figures 7 and 8, it is apparent that the system is slowly *escaping* the trapping zone over several market cycles as the oscillatory dynamics become more asymmetric; the system will eventually escape to an ESS if the attractor arc remains in the same position. We propose that policy influenced *arc tilt* and technology driven *arc drift* work to re-situate the arc and sustain a dynamic trapping zone. Therefore, oscillatory dynamics by market influence, arc tilt by policy developments and arc drift by technology adoption evolve in concert to maintain a perpetual pseudo-stable state.

Mathematically, our strategy payoffs when $n = 0$ ($a_{l0}$, $a_{f0}$, $d_{l0}$, $d_{f0}$) and our $\delta$ terms ($\delta^n$, $\delta^p$, $\delta^q$) exist as constants dependent on the labor economy of interest. With $n$, $p$ and $q$ as our environment variables, $A(n, p, q)$ and $B(n, p, q)$ denote payoffs for laborers and firms, respectively, where $n$, $p$ and $q$ are functions of time. An evolving orbit of forced dynamics can be explained through the confluence of $\dot{n}$, $\dot{p}$ and $\dot{q}$, which describe how each environment evolves over time. In addition to the particular examples demonstrated in

this work, in principle, we can characterize the subtle difference in terms of how specific assumptions about environment variables would impact gig labor preferences.

### 3.5.2. Implications for the Modern Gig Economy

In this work, we introduce three dynamics that help explain the present day co-existence of gig and employee strategies and why a single strategy has not dominated at an ESS. Market driven oscillatory dynamics reflect changes in labor preferences across economic conditions. Technology influenced arc drift illustrates the transformative nature of new technologies that increase opportunities for freelance labor and auxiliary work arrangements. Policy guided arc tilt reflects the regulation and transfer of economic risks between firms and laborers in an effort to maintain fair, sustainable labor economies. These three environments embrace in an interlocking dance, a balancing act, to create an evolving trapping zone where the labor economy can support the co-existence of labor strategies.

The described treble of dynamics, guided by $\dot{n}$, $\dot{p}$ and $\dot{q}$, serve as a checks-and-balances apparatus to trap the labor economy in a pseudo-stable state of strategy co-existence. When there is an irregular market cycle, such as a prolonged economic recession, it is sensible to expect new policies for labor protection and new opportunities for technology enterprise, both of which respond to balance out the market abnormality. When new technologies emerge, policy serves as the regulatory instrument to facilitate the prudent adoption of novel applications, and the market evolves to reflect future sentiment of emerging tech-enabled opportunities. When regulatory bodies introduce new labor policies, the market reacts to reflect sentiments regarding enterprise outlook, and new technologies emerge in response to evolving governance guidelines. This treble of dynamics is a *robust* balancing act; for example, not all policies are well designed, yet gig and employee strategies co-exist today, implying that technology and markets appropriately responded from the genesis of labor markets up to the present day. When one environment evolves abnormally, the other two environments must respond to force a persistent trapping zone.

While this treble of dynamics has evolved in concert to prolong the labor market's pseudo-stable state from the genesis of labor economies to the present day, in the present work, we do not claim that this evolving orbit of forced dynamics is an eternal stable equilibrium. Assuming a finite temporal interval from the genesis of labor markets to the present day, it can be inferred that this dynamical balancing act has been robust enough to weather centuries of irregular market events, poorly-designed policies and revolutionary technologies. Although robust, this balancing act may not be eternally resistant. Regarding the future of the gig economy, it is plausible that if one policy, market or technology development emerges and is so radical, the other environments may not be able to appropriately respond in time. Therefore, some labor economies, whether it be that of a specific sector or a single firm, may eventually break out of the pseudo-stable trapping zone and escape to an ESS where either only gig or only employee strategies exist.

In the present work, we establish a model to describe policy, technology and market influences through a treble of dynamics that support the co-existence of gig and employee strategies. While we presume these dynamics are robust given present-day strategy co-existence and the extensive history of labor markets, we leave future scenarios open to the possibility that some labor economies may shift entirely to one type of workforce. Given new technology, policy or market developments, it may be sensible for some industries to adopt a single labor strategy, while others may continue to champion a mixed strategy workforce. Thus, it is incumbent on the innovators, the policy makers, the laborers and the firms to appropriately respond to new environmental events in order to ensure the sustainable, fair futures of their respective labor economies.

## 4. Discussion

The emergence of the modern gig economy introduces a new set of employment considerations for firms and laborers. Among manifold regards, firms must elect between hiring a gig worker or an employee while balancing labor costs with product quality

and worker reliability. When deciding to participate in the gig economy, laborers must evaluate autonomy at the expense of financial stability and labor protections conferred with employee status. In practice, these elements of employment incentives and deterrents can be modeled with strategy-dependent payoffs, presenting a suitable opportunity for a game theoretical exploration. Influenced by several macroeconomic forces, these employment incentives are shaped by the nexus between dynamic market, technology and policy developments. On one hand, a bear market can discount worker-autonomy and accessible service demand from consumers. On the other, a bull market can enable workers to engage in a broader scope of alternative engagements and earn additional bonuses. Indeed, high and low skill laborers are impacted differently and have idiosyncratic susceptibilities to market changes. Regarding regulatory structures, policy behaves as a mechanism that transfers economic burdens between firms and laborers. For researchers and policy markers alike, there remains an unanswered question as to whether or how to regulate the gig economy. Adjacently, advancements in technology—in particular, digital platforms—have often been attributed as catalysts of growth for the modern gig economy. Contrarily, other technologies such as AI may implicate a future contraction of the servicing gig economy. Consolidating a multitude of micro and macro determinants, we explore how the compositions of firm and laborer strategies for gig or employee labor evolve under different market conditions, regulatory ordinances and technological developments.

In our work, we apply a game theoretical approach to study the evolution of strategy densities in firm and laborer populations, recasting employment incentives into strategy-dependent payoffs and fluctuating market conditions into an evolving environment variable. Formally, we extend the replicator equation to model oscillating dynamics in two-player asymmetric bi-matrix games with a time-evolving environment. While classical game theory centers on stable equilibrium solutions, we demonstrate a pseudo-stable state in which the system oscillates in a trapping zone orbit as a result of dynamic payoffs governed by three evolving environments. We extend our model to exhibit how changes in payoffs can transform the orientation and position of the system's oscillatory orbit, concepts we refer to as arc drift and arc tilt. Applying these concepts to our study of the gig economy, we demonstrate how technology and policy can implicate arc drift and tilt.

In the present work, we present three noteworthy contributions to existing scholarship on the gig economy and evolutionary dynamics. First, we extend the replicator equation to a new form of game, oscillating replicator dynamics with attractor arcs, introducing concepts of the attractor arc, driven oscillation, trapping zone and escape. We extensively study the behavior of a pseudo-stable equilibrium which is governed by evolving environments. We detail this pseudo-stable equilibrium with the notion of a trapping zone. The formalization of the attractor arc presents a novel analytical approach for evolutionary game theory where we are able to study new dynamics arising from arc transformations, which we formalize in our arc drift and arc tilt concepts. Further, we establish how escape can be bounded for extended periods through a stabilizing treble of dynamics: driven oscillation, arc drift and arc tilt.

Second, we present a generalized economic model to study strategy evolution in any labor economy, whether it be that of a country or a single firm, and demonstrate how market, technology and policy influence labor strategies in three distinct dynamics. We explore how evolving market conditions implicate different oscillatory dynamics based on how firms and laborers adjust their strategy payoff considerations in bear and bull markets. Next, we present a payoff framework to analyze the role of technology in the growth of the gig economy, informing tensions regarding the future of this new employment category. By exploring the nature of attractor arc drift, we establish payoff operations that imply the growth or contraction of the gig economy. We provide analysis that suggests technology, namely digital platforms, enabled low skill workers to sustainably participate in the gig economy, resulting in its neoteric rise. In our theoretical extension, we offer arguments that suggest the gig economy may either continue to grow or contract in the future. The direction of future gig economy growth depends on various technological

developments and a potential future re-constitution of work and enterprise. Finally, we explore regulatory implications within the gig economy, demonstrating how policy acts as a mechanism to transfer risk and economic burden between firms and laborers. In our model, we investigate the impact of shifting payoff utility between firms and laborers, which is reflected in an arc tilt transformation. We find that intervals of lenient and strict regulatory ordinances alter firm and worker sensitivities to different labor strategies.

Third, we establish an aggregate theory that addresses the treble of dynamics under technology, policy and market influence. We explore how the three dynamics work together to "trap" the system in a pseudo-stable equilibrium which manifests as an evolving orbit of forced dynamics. When there is a market abnormality, new policies and technologies are introduced to facilitate economic balance. When a radical policy is ratified, technology adapts to new governance guidelines, and markets reflect future enterprise outlook. When a transformative technology emerges, policy facilitates its cautious adoption and markets reflect future technology-enabled opportunities. While we presume that these three dynamics serve as a robust balancing apparatus that has historically helped sustain gig and employee strategy co-existence, we leave the future of some labor economies open to the possibility of single strategy dominance.

This work is founded on assumptions contingent on a number of limitations. We present our model's constraints, mapping out directions with promising opportunity for future research.

As discussed, our evolutionary model presents a pseudo-stable equilibrium conditional on the relationship between selection intensity, $\omega$, and the rates of environment evolution, $\dot{n}$, $\dot{p}$ and $\dot{q}$. Further research can be conducted to mathematically formalize trapping boundaries and escape velocities. Moreover, we also hypothesize that there exist regions in some systems wherein infinite oscillation in a trapping zone is possible; research on the alignment of attractor arcs and system symmetries may elucidate on this hypothesis.

In conclusion, we propose a model that incorporates a co-evolving treble of macro forces—markets, technology and policy—and demonstrate their respective influences on labor strategies in the gig economy. We demonstrate how technology is a driver of change in the labor economy and how policy is integral to the sustainability of new systems and the protection of involved parties. The primary goals of this paper were the further comprehension of micro and macro influences on firm and laborer incentives for gig adoption. We provide researchers, policy makers and industrialists alike with a novel evolutionary model and payoff framework approach for better understanding firm and laborer behaviors in the gig economy. Finally, it is our hope that scholarship on the gig economy can extend to study adjacent topics of education and economic mobility. Perhaps the rise of distributed and widely-accessible education resources paired with a reconstitution of work and enterprise will establish the future gig economy as a means of economic mobility.

**Supplementary Materials:** The following are available online at https://www.mdpi.com/article/10.3390/g12020049/s1, Figure S1: Evolution of Strategy Densities for Small Low-Skill Firm with Initial Conditions $(0.4417, 0.5554)$, the attractor position at $n = 0.5$, an approximation for a point in the trapping zone; $\omega = 0.00000001$; and Payoff Matrices Small Low Bear and Small Low Bull, see Tables S1 and S2. (a) Trapping Zone Orbit (b) Attractor Arc (c) Labor Strategy Oscillation Over Three Market Cycles. In (a), the trapping zone orbit is plotted in yellow, and attractor positions at $n = 0$ and $n = 1$ are represented in orange and blue respectively. In (b), we plot a reference attractor arc in purple with attractor positions when $n = 0$, $n = 0.25$, $n = 0.5$, $n = 0.75$ and $n = 1$. (c) visualizes the fluctuation in firm and laborer preferences for gig strategies over three market cycles, Figure S2: Evolution of Strategy Densities for Large Low-Skill Firm with Initial Conditions $(0.5186, 0.5535)$, the attractor position at $n = 0.5$, an approximation for a point in the trapping zone; $\omega = 0.0000000002$; and Payoff Matrices Large Low Bear and Large Low Bull, see Tables S3 and S4 (a) Trapping Zone Orbit (b) Attractor Arc (c) Labor Strategy Oscillation Over Three Market Cycles. In (a), the trapping zone orbit is plotted in yellow, and attractor positions at $n = 0$ and $n = 1$ are represented in orange and blue respectively. In (b), we plot a reference attractor arc in purple with attractor positions when

$n = 0$, $n = 0.25$, $n = 0.5$, $n = 0.75$ and $n = 1$. (c) visualizes the fluctuation in firm and laborer preferences for gig strategies over three market cycles, Figure S3: Evolution of Strategy Densities for Small High-Skill Firm with Initial Conditions $(0.5498, 0.4298)$, the attractor position at $n = 0.5$, an approximation for a point in the trapping zone; $\omega = 0.00000001$; and Payoff Matrices Small High Bear and Small High Bull, see Table S5 and S6 (a) Trapping Zone Orbit (b) Attractor Arc (c) Labor Strategy Oscillation Over Three Market Cycles. In (a), the trapping zone orbit is plotted in yellow, and attractor positions at $n = 0$ and $n = 1$ are represented in orange and blue respectively. In (b), we plot a reference attractor arc in purple with attractor positions when $n = 0$, $n = 0.25$, $n = 0.5$, $n = 0.75$ and $n = 1$. (c) visualizes the fluctuation in firm and laborer preferences for gig strategies over three market cycles, Figure S4: Evolution of Strategy Densities for Large High-Skill Firm with Initial Conditions $(0.5973, 0.4302)$, the attractor position at $n = 0.5$, an approximation for a point in the trapping zone; $\omega = 0.0000000002$; and Payoff Matrices Large High Bear and Large High Bull, see Tables S7 and S8 (a) Trapping Zone Orbit (b) Attractor Arc (c) Labor Strategy Oscillation Over Three Market Cycles. In (a), the trapping zone orbit is plotted in yellow, and attractor positions at $n = 0$ and $n = 1$ are represented in orange and blue respectively. In (b), we plot a reference attractor arc in purple with attractor positions when $n = 0$, $n = 0.25$, $n = 0.5$, $n = 0.75$ and $n = 1$. (c) visualizes the fluctuation in firm and laborer preferences for gig strategies over three market cycles, Table S1: GameState Contract Demand Distribution, Table S2: GameState Payoff Coefficients, Table S3: $4 \times 4$ Payoff Matrix: Matching Strategy Pairs, Table S4: $4 \times 2$ Payoff Matrix: Matching Strategy Pairs for High-Skill Contracts, Table S5: $4 \times 2$ Payoff Matrix: Matching Strategy Pairs, Table S6: $4 \times 2$ Payoff Matrix: Matching Strategy Pairs for Firms, Table S7: $4 \times 2$ Payoff Matrix: Mismatching Strategy Pairs for Firms, Table S8: $4 \times 2$ Payoff Matrix: Matching Strategy Pairs for Laborers, Table S9: $4 \times 2$ Payoff Matrix: Mismatching Strategy Pairs for Laborers, Table S10: $4 \times 4$ Payoff Matrix, Table S11: $2 \times 2$ Payoff Matrix, Table S12: $2 \times 2$ Payoff Matrix for GameState in Setting Small Low Bear, Table S13: $2 \times 2$ Payoff Matrix for GameState in Setting Small Low Bull, Table S14: $2 \times 2$ Payoff Matrix for GameState in Setting Large Low Bear, Table S15: $2 \times 2$ Payoff Matrix for GameState in Setting Large Low Bull, Table S16: $2 \times 2$ Payoff Matrix for GameState in Setting Small High Bear, Table S17: $2 \times 2$ Payoff Matrix for GameState in Setting Small High Bull, Table S18: $2 \times 2$ Payoff Matrix for GameState in Setting Large High Bear, Table S19: $2 \times 2$ Payoff Matrix for GameState in Setting Large High Bull.

**Author Contributions:** Conceptualization, K.H. and F.F.; Formal analysis, K.H. and F.F.; Funding acquisition, F.F.; Investigation, K.H. and F.F.; Methodology, K.H. and F.F.; Project administration, F.F.; Supervision, F.F.; Writing—original draft, K.H.; Writing—review & editing, K.H. and F.F. All authors have read and agreed to the published version of the manuscript.

**Funding:** We are grateful for support from the Bill and Melinda Gates Foundation (award no. OPP1217336), the NIH COBRE Program (grant no.1P20GM130454), the Neukom CompX Faculty Grant, the Dartmouth Faculty Startup Fund and the Walter and Constance Burke Research Initiation Award.

**Institutional Review Board Statement:** Not Applicable.

**Informed Consent Statement:** Not Applicable.

**Data Availability Statement:** All data have been included in the paper.

**Acknowledgments:** K.H. would like to thank Xin Wang for discussions which helped improve the work.

**Conflicts of Interest:** The authors declare no conflict of interest. The funders had no role in the design of the study; in the collection, analyses, or interpretation of data; in the writing of the manuscript, and in the decision to publish the results.

## Appendix A. Evolutionary Model

*Appendix A.1. System Equilibria*

In this section, we solve for our evolutionary system's fixed points for the general case. For each fixed point, we analyze the stability of the equilibrium and offer an explanation.

Appendix A.1.1. Fixed Points

Solving our system of two equations and two unknowns, we reach a general solution set that contains five fixed points. Below, we list each fixed point in the form $(x_1, y_1)^*$.

$FixedPoint_1 = (0,0)^*$

$FixedPoint_2 = (1,0)^*$

$FixedPoint_3 = (0,1)^*$

$FixedPoint_4 = (1,1)^*$

$FixedPoint_5 = (\dfrac{-(c_{f0}-d_{f0}-c_{f0}n+c_{f1}n+d_{f0}n-d_{f1}n)}{(a_{f0}-b_{f0}-c_{f0}+d_{f0}-a_{f0}n+a_{f1}n+b_{f0}n-b_{f1}n+c_{f0}n-c_{f1}n-d_{f0}n+d_{f1}n)},$

$\dfrac{-(b_{l0}-d_{l0}-b_{l0}n+b_{l1}n+d_{l0}n-d_{l1}n)}{(a_{l0}-b_{l0}-c_{l0}+d_{l0}-a_{l0}n+a_{l1}n+b_{l0}n-b_{l1}n+c_{l0}n-c_{l1}n-d_{l0}n+d_{l1}n)})^*$

Fixed points 1, 2, 3 and 4 lie on the extremes of our system, and $FixedPoint_5$ is our only internal equilibrium. In our model, this implies that $FixedPoint_5$ is the only equilibrium with a co-existence of gig workers and employees.

Appendix A.1.2. Stability Analysis

To analyze the stability of each equilibrium, we examine the eigenvalues of the Jacobian matrix for each fixed point. For a fixed point to be asymptotically stable, eigenvalues of the Jacobian must have all negative real parts. If eigenvalues have all positive real parts, the fixed point is unstable. If the set of eigenvalues includes both positive and negative real parts, the equilibrium is a saddle point.

In order to feasibly analyze the negativity of eigenvalues, we reduce the number of generalized parameters. For simplification purposes, we assign all mismatching strategies a payoff of 0 as no mutual labor agreement is made between firm and laborer. To further simplify, we demonstrate our stability analysis with $n = 0$ rather than allowing n to remain a generalized parameter. The remaining parameters are matching strategy payoffs; for all GameStates, these payoffs take positive values. Given these assumptions, the simplified Jacobian is as follows.

$$J = \begin{bmatrix} -(2x_1-1)(a_{l0}y_1-d_{l0}+d_{l0}y_1) & -x_1(a_{l0}+d_{l0})(x_1-1) \\ -y_1(a_{f0}+d_{f0})(y_1-1) & -(2y_1-1)(a_{f0}x_1-d_{f0}+d_{f0}x_1) \end{bmatrix}_{(x_1^*,y_1^*)} \quad \text{(A1)}$$

**Saddle Points**

We find that our internal equilibrium $FixedPoint_5$ is a saddle point. The set of eigenvalues always takes both positive and negative values as the two eigenvalues are opposites of each other.

$Eigenvalue_1 = \dfrac{(a_{f0}a_{l0}d_{f0}d_{l0}(a_{f0}+d_{f0})(a_{l0}+d_{l0}))^{(1/2)}}{(a_{f0}a_{l0}+a_{f0}d_{l0}+a_{l0}d_{f0}+d_{f0}d_{l0})}$

$Eigenvalue_2 = -\dfrac{(a_{f0}a_{l0}d_{f0}d_{l0}(a_{f0}+d_{f0})(a_{l0}+d_{l0}))^{(1/2)}}{(a_{f0}a_{l0}+a_{f0}d_{l0}+a_{l0}d_{f0}+d_{f0}d_{l0})}$

**Unstable Fixed Points**

We find that $FixedPoint_2$ $(1,0)^*$ and $FixedPoint_3$ $(0,1)^*$, equilibria at mismatching extremes, are unstable. Since all matching strategy payoffs are positive, both eigenvalues of the Jacobian matrix for each of the two fixed points are always positive.

For $FixedPoint_2$, $Eigenvalue_1 = a_{f0}$ and $Eigenvalue_2 = d_{l0}$.
For $FixedPoint_3$, $Eigenvalue_1 = a_{l0}$ and $Eigenvalue_2 = d_{f0}$.

If the system begins at one of these unstable fixed points, the system will not remain stationary. Rather, the system will evolve on a trajectory towards a stable fixed point.

**Stable Fixed Points**

We find that $FixedPoint_1$ $(0,0)^*$ and $FixedPoint_4$ $(1,1)^*$, equilibria at matching extremes, are unstable. Since all matching strategy payoffs are positive, both eigenvalues of the Jacobian matrix for each of the two fixed points are always negative.

For $FixedPoint_1$, $Eigenvalue_1 = -d_{f0}$ and $Eigenvalue_2 = -d_{l0}$.

For $FixedPoint_4$, $Eigenvalue_1 = -a_{f0}$ and $Eigenvalue_2 = -a_{l0}$.

If either of the initial condition begins at, or the system evolves to, one of these stable fixed points, the system will remain stationary. These two stable fixed points are our evolutionary stable strategies (ESS). At $(0,0)^*$, firms and laborers both have a density of 0 for the gig strategy, implying that both populations consist entirely of employee strategies. At $(1,1)^*$, firm and laborer populations are fully dominated by gig strategies. If the system evolves to an ESS, no auxiliary strategies will be able to invade the dominating strategy population given an initially low strategy density [33,44]. In other words, if the labor market evolves to a stage where both laborers and firms consist entirely of gig strategies, the system will forever remain fixed, implying that gig workers will dominate the labor market forever and that there will never exist an employee strategy again. Likewise, if the labor market evolves to a stage where both laborers and firms are comprised entirely of employee strategies, the system will remain fixed and employee strategies will dominate the labor market forever.

Appendix A.1.3. Concept Visuals

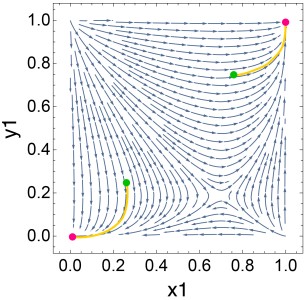

**Figure A1.** Evolutionary behavior for $n = 0$, $\dot{n} = 0$ with theoretical GameState payoff, see Figure 1. In this visualization, green represents initial condition, yellow represents the evolutionary path and red represents the final system position at an ESS. Two evolutionary journeys are visualized with initial conditions $(\frac{1}{4}, \frac{1}{4})$ and $(\frac{3}{4}, \frac{3}{4})$.

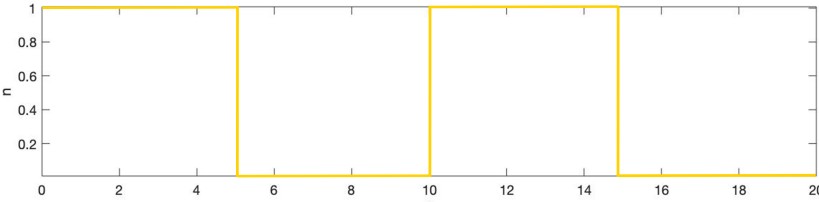

**Figure A2.** The chosen $\dot{n}$, n as a Function of t, to be used in the demonstrations.

*Appendix A.2. Oscillating Replicator Dynamics*

Appendix A.2.1. Computational Notes

We generate our evolutionary diagrams with Matlab and our phase diagrams with Mathematica. We also employ Matlab for calculating attractor arc reference points, fixed points, the Jacobian, eigenvalues and streamplot equations. We use the Adobe Photoshop editor for superimposing diagrams and incorporating additional visual aids.

Appendix A.2.2. Trapping Zone Orbit

We select our initial condition to be $(0.45, 0.40)$, the attractor position at $n = 0.5$, an approximation for a point in the trapping zone. This selection implies that we assume our system has previously oscillated in the trapping zone up until this moment in time. This assumption is sensible because the labor market maintains a co-existence of gig workers and employees.

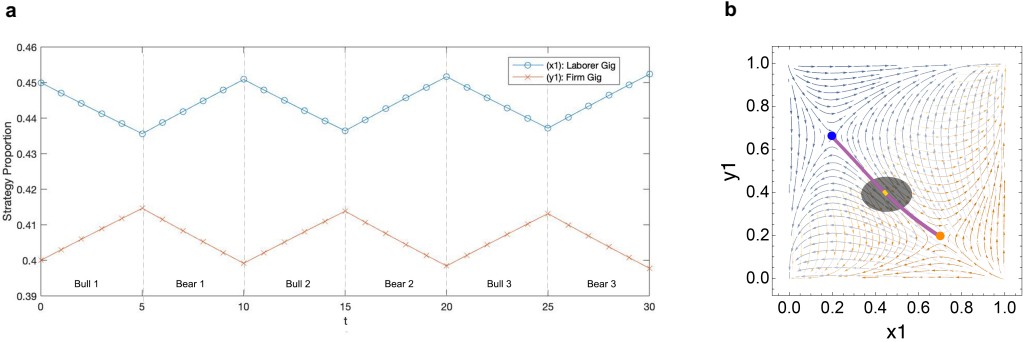

**Figure A3.** Trapping zone oscillation with initial conditions $(0.45, 0.40)$, $\omega = 0.005$ and $n = 1$ and theoretical GameState pair, see Figure 1. (**a**) Mismatching oscillatory behavior in trapping zone. (**b**) Trapping zone orbit. We illustrate the trapping zone orbit in yellow. A reference attractor arc is plotted in purple and attractor positions at $n = 0$ and $n = 1$ are represented in orange and blue, respectively. The opaque black ellipse is a background element for visual contrast.

Appendix A.2.3. Escape Demonstration with Different Initial Conditions

In this section, we illustrate an example of *escape* by increasing $\omega$ by a factor of 20, $\omega = 0.1$, such that the system reaches escape boundary. In Figure A4, the increase in $\omega$ occurs at the start of the bull market. In Figure A5, the increase in $\omega$ occurs at the start of the bear market. The purpose of the following demonstration is to illustrate how initial conditions can alter escape destination. Therefore, claims founded on escape destination are indefensible because escape destination is determined by arbitrary preparations of initial conditions.

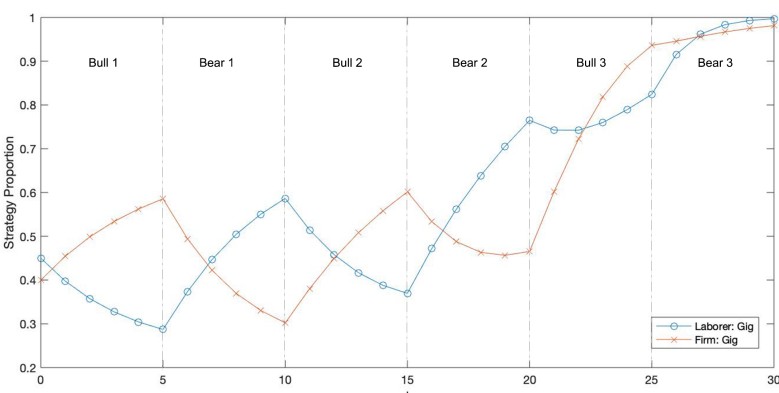

**Figure A4.** Escape demonstration with initial conditions $(0.45, 0.40)$, $\omega = 0.1$ and $n = 1$ and theoretical GameState pair, see Figure 1.

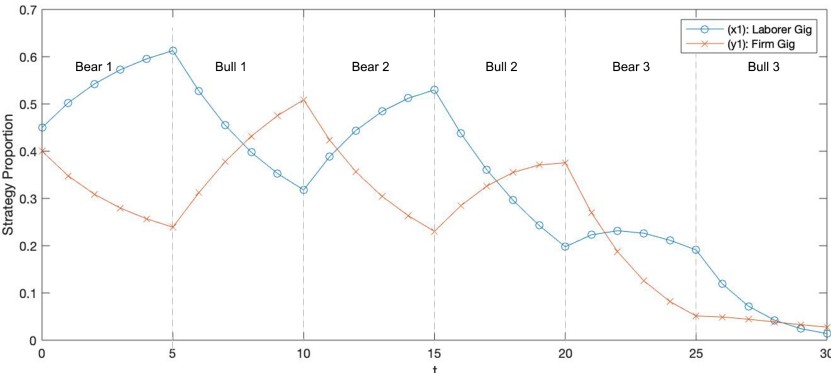

**Figure A5.** Escape demonstration with initial conditions $(0.45, 0.40)$, $\omega = 0.1$ and $n = 0$ and theoretical GameState pair, see Figure 1.

## Appendix B. Payoff Matrices

**a**

| $_{Laborer}\backslash^{Firm}$ | Gig | Employee |
|---|---|---|
| Gig | [9, 7] | [0, 0] |
| Employee | [0, 0] | [2, 3] |

**b**

| $_{Laborer}\backslash^{Firm}$ | Gig | Employee |
|---|---|---|
| Gig | [3, 2] | [0, 0] |
| Employee | [0, 0] | [6, 8] |

**Figure A6.** Theoretical GameState payoff No. 2: (**a**) Bear market, $n = 0$; (**b**) Bull market, $n = 1$.

**a**

| $_{Laborer}\backslash^{Firm}$ | Gig | Employee |
|---|---|---|
| Gig | [9, 7] | [0, 0] |
| Employee | [0, 0] | [2, 7] |

**b**

| $_{Laborer}\backslash^{Firm}$ | Gig | Employee |
|---|---|---|
| Gig | [3, 8] | [0, 0] |
| Employee | [0, 0] | [6, 8] |

**Figure A7.** Payoffs for vertical attractor arc demonstration. (**a**) Bear market, $n = 0$; (**b**) Bull market, $n = 1$.

**a**

| $_{Laborer}\backslash^{Firm}$ | Gig | Employee |
|---|---|---|
| Gig | [2, 7] | [0, 0] |
| Employee | [0, 0] | [2, 3] |

**b**

| $_{Laborer}\backslash^{Firm}$ | Gig | Employee |
|---|---|---|
| Gig | [3, 2] | [0, 0] |
| Employee | [0, 0] | [3, 8] |

**Figure A8.** Payoffs for horizontal attractor arc demonstration. (**a**) Bear market, $n = 0$; (**b**) Bull market, $n = 1$.

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
