# Peer review of "Evolutionary Dynamics of Gig Economy Labor Strategies under Technology, Policy and Market Influence"

_games, doi:10.3390/g12020049_

Round 1

Reviewer 1 Report

The authors honestly respond to comments and made the necessary corrections

Author Response

We are pleased to hear that the changes made in our last revision addressed the referee’s comments and questions. Again, we are very grateful to the referee for their helpful comments which helped us greatly improve the paper.

Reviewer 2 Report

You have not followed my comments. Hence, read again my recommendations:

The paper is a very important contribution to the applied economics. The paper is well-written and has a plausible structure. The underling methodology is fine. The authors should at least cursorily mention the merits of differential game theory approach, which could also be employed in the context of the present inquiry. To this end refer to the following applied research Distributional Bargaining and the Speed of Structural Change in the Petroleum Exporting Labor Surplus Economies | SpringerLink

Author Response

We thank the referee for their careful review of our work. As shown below, we have briefly mentioned the merits of employing a differential game theory approach in the context of the present inquiry and have highlighted the suggested paper as cited in Ref. [52]. We hope that we have revised the manuscript thoroughly to your satisfaction.

We are very grateful to the referee for this endorsement of our work. In our revision, we mention the merits of differential game theory approach, its relation to evolutionary game theory and its application in the study of labor economies. (p.3, lines 100-108) (p. 3, lines 126-128)

“Pioneered by John von Neumann \cite{kuhn1958john}, the study of modern game theory anchors itself in the assumption that players make rational decisions based on the respective payoff incentives conferred with each strategy \cite{smith1973logic, nash1950bargaining, weitz2016oscillating}. Although modern game theory initially focused on static theories, differential game theory, a subject introduced by Rufus Isaacs, considers the state of players with time as a continuous variable \cite{Quincampoix2012}. While classical game theory developed to address questions in economics~\citep{nash1950bargaining, von2007theory}, the field of evolutionary game theory, a theoretical extension that models how populations change strategies over time~\citep{cressman2014replicator}, finds its roots in biology~\citep{smith1973logic, cressman2014replicator}. Since its inception in 1973~\citep{smith1973logic}, evolutionary game theory has broadened in application beyond its early biological origins to study social interactions and population behaviors across various academic fields~\citep{cressman2014replicator, alexander2002evolutionary,bear2016intuition,rand2011evolution,apicella2019evolution,perc2017statistical}.” (p.3, lines 100-108)

Economic behavior in the labor economy has been investigated through the lens of both evolutionary game theory and differential game theory \citep{sadik2020distributional,carrera2020co,leitmann1974differential,araujo2010evolutionary}. For instance, Sadik-Zada derived feasible bargaining equilibria including the antagonistic and the allocation modes, based on a noncooperative differential game model~\citep{sadik2020distributional}. (p. 3, lines 126-129)

Reviewer 3 Report

The authors put in quite a bit of effort to revising their paper given the limited time they had. 

Nonetheless, I still feel that some of my concerns remain unaddressed. I shall raise the central point first: 

In their letter, the authors have reiterated that prior to digital platforms, gig-employment was undertaken only by those with the highest market standing ("An angel investor or company adviser or perhaps even a McKinsey consultant"). Regardless of what the authors have written in their letter, I remain unconvinced. Indeed, for example, Bidwell & Briscoe (2009) (AMJ), which the authors cite, highlights that the decision to work as an independent contractor is polarised. 

Highly experienced contractors have _chosen_ nonstandard work, while inexperienced contractors have few outside options outside nonstandard work. This push/pull tension is missing from the authors' discussion. 

I do not want to get into a debate on the essence of the historical gig economy. Instead, I would like to suggest a different framing. Instead of holding on to the term "gig work," which leads to confusion, the authors could concentrate on "nonstandard work arrangements. For example, most taxi drivers in large US cities have always been independent contractors. Their entry to the market was still regulated by taxi medallions, which led to fairly good working conditions. The advent of digital taxi platforms introduced much more competition among taxi drivers (and an increase in nonstandard work), which resulted in lower earnings and poorer working conditions. Now that many cities are trying to regulate Uber and Lyft more, the scales might start tilting to the other direction. This tilt has included less free entry to the taxi driver labour market and slightly higher earnings. 

If I understand correctly, this is the oscillation between high and low pay nonstandard work that the authors' model captures? Obviously, this gradual shift is moderated by the overall demand for labour ('bear and bull cycles').

Thus, my suggestion is to cut out most of the stuff on mckinsey consultants and advisors. Instead, frame the discussion about nonstandard work contracts (of which the digital gig economy is one example). I do not think this affects the essence of the argument much. 

The second point I want to reiterate: 

The authors should map the novel concepts (attactor arc, trapping zone, pseudo-stable state) to what they imply in the labour market context. The authors should do this already in the intro.

Third: the authors really should have the language checked. The article is hard to follow and sometimes unintendedly funny ('cannons of evolutionary dynamics'). I believe that the authors mean labor markets when they discuss labor economies, etc. 

Author Response

  • The authors put in quite a bit of effort to revising their paper given the limited time they had. 

We thank the referee for their careful reading of our work. We believe that their detailed comments in the review process helped us to greatly improve the paper.

  • Nonetheless, I still feel that some of my concerns remain unaddressed. I shall raise the central point first: 

In their letter, the authors have reiterated that prior to digital platforms, gig-employment was undertaken only by those with the highest market standing ("An angel investor or company adviser or perhaps even a McKinsey consultant"). Regardless of what the authors have written in their letter, I remain unconvinced. Indeed, for example, Bidwell & Briscoe (2009) (AMJ), which the authors cite, highlights that the decision to work as an independent contractor is polarised. 

Highly experienced contractors have _chosen_ nonstandard work, while inexperienced contractors have few outside options outside nonstandard work. This push/pull tension is missing from the authors' discussion. 

I do not want to get into a debate on the essence of the historical gig economy. Instead, I would like to suggest a different framing. Instead of holding on to the term "gig work," which leads to confusion, the authors could concentrate on "nonstandard work arrangements. For example, most taxi drivers in large US cities have always been independent contractors. Their entry to the market was still regulated by taxi medallions, which led to fairly good working conditions. The advent of digital taxi platforms introduced much more competition among taxi drivers (and an increase in nonstandard work), which resulted in lower earnings and poorer working conditions. Now that many cities are trying to regulate Uber and Lyft more, the scales might start tilting to the other direction. This tilt has included less free entry to the taxi driver labour market and slightly higher earnings. 

If I understand correctly, this is the oscillation between high and low pay nonstandard work that the authors' model captures? Obviously, this gradual shift is moderated by the overall demand for labour ('bear and bull cycles').

Thus, my suggestion is to cut out most of the stuff on mckinsey consultants and advisors. Instead, frame the discussion about nonstandard work contracts (of which the digital gig economy is one example). I do not think this affects the essence of the argument much. 

We thank the referee for this detailed comment. We have implemented the reviewer’s suggestions and reframed our discussion to focus on nonstandard work arrangements. Additionally, we have introduced discussion around differences in high and low skill laborers pursuing nonstandard work arrangements. Per the reviewer’s suggestion, we have also cut out the content discussing McKinsey consultants and company advisors. As the reviewer points out, we agree that this reframing does not affect the essence of the argument much. (p. 15-16, lines 443-449)

“Our model suggests that the premature gig economy (see Figure 13a), which consisted of fewer workers pursuing nonstandard or gig employment, involved gig labor that averaged higher payoffs compared to employee labor (see Figure 11). Since payoff is determined by compensation, high payoff implies high compensation. Appropriately, when gig payoffs are very high, each company can afford to hire a small amount of these elite, skilled nonstandard workers. This explains why the attractor arc for high gig payoffs is located near $(0,0)^*$, indicating a labor composition consisting of few gig workers. Often, firms recruit highly skilled nonstandard workers to provide rare, distinctive skills and abilities that are unavailable within the company \cite{kalleberg2003externalizing}. Previous studies suggest that highly skilled workers benefit more from the flexibility provided by nonstandard employment and possess increased negotiating leverage due to their desirable skill-sets \cite{golden2008limited, kalleberg2003externalizing}. Highly skilled workers may be more inclined to pursue nonstandard employment due to this increased flexibility \cite{barley2011gurus}. Contrarily, low-skill workers pursuing nonstandard employment are more exposed to the precarity of evolving market conditions and possess less bargaining power \cite {kalleberg2000bad, catanzarite2000brown}. 

More recently, technology has enabled firms to outsource more piecemeal work to low-skill workers. Existing literature suggests that technology reduces task complexity, enabling workers with fewer skills to complete these jobs \cite{smith1997new, smith1998fractured}. Without contemporary gig platforms, there was a higher barrier to discover and secure auxiliary work arrangements. Digital platforms enable low-skill workers to feasibly participate in the gig economy by coordinating and matching streams of labor supply and demand and providing workers with a sustained cascade of work opportunity. Examples of such technologies include ride-sharing and last-mile delivery apps such as Uber, Lyft and DoorDash as well as freelancing websites such as Upwork, all of which introduce mostly low-skill, low-payoff workers to the gig economy. While some readers may regard sharing-economy applications as platform operations, drivers as gig workers and customers as the firm who employs the driver, we clarify our interpretation of the sharing-economy platform as the firm. As low-skill gig workers such as Uber drivers flooded the gig economy, gig payoffs decreased relative to employee payoffs. This is consistent with our model as the attractor arc shifts towards $(1,1)^*$ from the blue to yellow arc during this development, reflecting the neoteric growth of the modern gig economy.”

  • The second point I want to reiterate: 

The authors should map the novel concepts (attactor arc, trapping zone, pseudo-stable state) to what they imply in the labour market context. The authors should do this already in the intro.

We thank the referee for this suggestion. We have followed this suggestion by adding a brief elaboration in the end of the Introduction part to explicitly point out these concepts and their implications in the context of labor market will be used to demonstrate the impacts of technology, policy and market on influence gig economy preference. The revised part now reads (p. 4 lines148-155):

“In this work, we provide researchers, policy makers and industrialists alike with a proof-of-concept evolutionary game theory approach along with an environment-dependent payoff generation framework for better understanding firm and laborer behaviors in the gig economy. Our modeling framework and the concepts including attractor arc, trapping zone, and pseudo-stable state can be used to demonstrate how technology is a driver of change in the labor economy and how policy is integral to the sustainability of new systems and the protection of involved parties. The present paper aims to further comprehension of micro and macro influences on firm and laborer incentives for gig adoption in these regards.”

  • Third: the authors really should have the language checked. The article is hard to follow and sometimes unintendedly funny ('cannons of evolutionary dynamics'). I believe that the authors mean labor markets when they discuss labor economies, etc. 

We apologize for the confusion that our original wording may have caused. We have carefully streamlined our presentation throughout the manuscript and also have attempted to clarify any points of confusions in our revisions (see appended pdf with all changes highlighted). For example, we have corrected the phrase “cannons of evolutionary dynamics” as pointed out by the reviewer, and changed the sentence to read as follows:

“In this paper, we impart three notable contributions to the field of evolutionary dynamics and existing literature on gig economy (and more broadly nonstandard work arrangements). (p. 3, lines 135-136)

Round 2

Reviewer 3 Report

The authors have addressed my critiques sufficiently. 

This manuscript is a resubmission of an earlier submission. The following is a list of the peer review reports and author responses from that submission.

Round 1

Reviewer 1 Report

The authors apply known methods of analysis of evolutionary games to the new economic reality. In this context, the work does not constitute a significant scientific discovery. Nevertheless, analyzes of this type are very interesting and can be inspiring.

I have a few comments on the work:

  • The authors write about ‘’new extension to the replicator equation’’. In my opinion, this is the standard equation. Authors should discuss this in the context of other works.
  • Tables at work are unsigned. Some of them contain incomprehensible elements. For example what does the word 'blue' mean in the tables on pages 27 and 28?
  • The second part of the job is very chaotic. I had big problems with understanding the authors' arguments. There are many bizarre equations at work such as (A8). Such a notation deviates from mathematical standards.

I think that the second part of the work (computational) the authors should think over and write more clearly. A non-specialist can get lost in it.

Author Response

  • The authors apply known methods of analysis of evolutionary games to the new economic reality. In this context, the work does not constitute a significant scientific discovery. Nevertheless, analyzes of this type are very interesting and can be inspiring.

We very much appreciate this criticism, and we have restructured our paper to better situate our core concepts in the context of the gig economy and existing scholarship. We have generalized our model to enable the study of all labor economies, in lieu of specific payoffs. In addition to presenting a generalized framework for individual dynamics as influenced by market, technology and policy changes, we now describe how these three environments work together to sustain a pseudo-stable equilibrium. While canonical studies of evolutionary game theory have focused on the evolutionary stable strategy, we believe that our study of a pseudo-stable state with attractor arcs (namely, oscillatory dynamics arising from market condition changes) is a novel contribution to the standard approach of replicator dynamics. We apologize for the lack of clarity in situating key concepts in our original manuscript. We have made all the suggested changes and have clarified any points of confusion.

On the economic front, while previous studies have explored the gig economy on the dimensions of technology, policy and markets individually, our paper presents new insights regarding the nexus between these three macro environments. Our study finds that these three environments work in concert and serve as a checks-and-balances apparatus to sustain a pseudo-stable co-existence of gig and employee strategies.

We greatly appreciate the reviewer’s enthusiasm for the inspirational impact that these types of analyses can present.

I have a few comments on the work:

  • The authors write about ‘’new extension to the replicator equation’’. In my opinion, this is the standard equation. Authors should discuss this in the context of other works.

We thank the reviewer for this suggestion.

In two new sections, (see section 3.2, p. 10-15, lines 337-403; and section 3.5, p. 20-22, lines 545-617), we have generalized our model and expanded the core concepts (attractor arc, driven oscillation, trapping zone, escape) in context of the labor economy. We believe that these new sections help demonstrate the novelty of our study of a sustaining pseudo-stable state as opposed to the conventional focus on ESS analysis. With our formalization of the attractor arc, we demonstrate how changing payoffs result in a variety of attractor arc transformations, presenting a novel analytical approach for evolutionary game theory.

Additionally, we have reworked our discussion to account for our new generalized model, which we believe also helps situate the novelty of our core concepts (section 4, lines 639-703).

We have reworded the sentence to read “new conceptual extensions to the replicator equation” to clarify that our contribution is a conceptual extension to the approach of replicator equation, rather than an extension of the equation itself  (p. 1, line 14, 122).

  • Tables at work are unsigned. Some of them contain incomprehensible elements. For example what does the word 'blue' mean in the tables on pages 27 and 28?

We apologize for the confusion. We have noted some Latex notation errors in the original manuscript that resulted in a few formatting errors. We have fixed all these errors to eliminate incomprehensible elements. We are grateful to the referee for a careful reading of our manuscript.

  • The second part of the job is very chaotic. I had big problems with understanding the authors' arguments. There are many bizarre equations at work such as (A8). Such a notation deviates from mathematical standards.

We sincerely apologize for the confusion that this section of the manuscript may have caused. In the revised manuscript, we have moved our sample payoff generation to a supplementary information file and fixed our notation. In place of this, we have added two new sections (see section 3.2, p. 10-15, lines 337-403; and section 3.5, p. 20-22, lines 545-617) and edited the technology (section 3.3, p. 15-17, lines 404-486) and policy (section 3.4, p. 17-20, lines 487-544) theoretical extensions to generalize our model to all labor economies. While we initially presented analysis based on our hypothetically generated payoffs, our new model enables the analysis of all labor economies, whether it be that of a country or a single firm, and is indiscriminate of the specific values of payoffs. In the revised manuscript, we reposition our findings and central analysis on the generalized model rather than specific payoffs.

  • I think that the second part of the work (computational) the authors should think over and write more clearly. A non-specialist can get lost in it.

Again, we sincerely apologize for the confusion that this section of our original manuscript may have caused. In our revision, we have centered our manuscript on a new generalized framework that can apply to all payoffs. Therefore, we have removed this “second part of the work” (where we computationally generated payoffs) from the primary manuscript. We thank the reviewer for thoughtful consideration for how our manuscript reads to broader audiences.

Reviewer 2 Report

The paper is a very important contribution to the applied economics. The paper is well-written and has a plausible structure. The underling methodology is fine. The authors should at least cursorily mention the merits of differential game theory approach, which could also be employed in the context of the present inquiry. To this end refer to the following applied research Distributional Bargaining and the Speed of Structural Change in the Petroleum Exporting Labor Surplus Economies | SpringerLink

Author Response

  • The paper is a very important contribution to the applied economics. The paper is well-written and has a plausible structure. The underling methodology is fine. The authors should at least cursorily mention the merits of differential game theory approach, which could also be employed in the context of the present inquiry. To this end refer to the following applied research Distributional Bargaining and the Speed of Structural Change in the Petroleum Exporting Labor Surplus Economies | SpringerLink

We are very grateful to the referee for this endorsement of our work. In our revision, we properly acknowledge this reference pointed out by the reviewer (see below), and also we highlight the merits of differential game theory approach, its relation to evolutionary game theory and its application in the study of labor economies. (p.3, lines 100-108) (p. 3, lines 126-127)

“Pioneered by John von Neumann \cite{kuhn1958john}, the study of modern game theory anchors itself in the assumption that players make rational decisions based on the respective payoff incentives conferred with each strategy \cite{smith1973logic, nash1950bargaining, weitz2016oscillating}. Although modern game theory initially focused on static theories, differential game theory, a subject introduced by Rufus Isaacs, considers the state of players with time as a continuous variable \cite{Quincampoix2012}. While classical game theory developed to address questions in economics~\citep{nash1950bargaining, von2007theory}, the field of evolutionary game theory, a theoretical extension that models how populations change strategies over time~\citep{cressman2014replicator}, finds its roots in biology~\citep{smith1973logic, cressman2014replicator}. Since its inception in 1973~\citep{smith1973logic}, evolutionary game theory has broadened in application beyond its early biological origins to study social interactions and population behaviors across various academic fields~\citep{cressman2014replicator, alexander2002evolutionary,bear2016intuition,rand2011evolution,apicella2019evolution,perc2017statistical}.” (p.3, lines 100-108)

“Scholars have examined economic behavior in the labor economy through the lens of both evolutionary game theory and differential game theory \cite{sadik2020distributional,carrera2020co,leitmann1974differential,araujo2010evolutionary}.”   (p. 3, lines 126-127)

Reviewer 3 Report

The article unfolds on two levels that remain quite separate throughout the text: the economic model and its mathematical analysis. Moreover, I find the treatment of both levels lacking.

On the economic side, the important part is the derivation of the interaction between firms are workers as a battle of sexes. The details, given in appendix A, rely on behavioral rules which I find rather arbitrary. The chosen mathematical models fit better with other social problems, the easiest being coordination on the same action/norm in a multi-cultural society.

On the mathematical side, the paper is, potentially, more appealing. Letting the payoff structure of the game change (albeit exogenously) is quite interesting. However, I could not find formal results, e.g. about convergence to an arc. Pictures look nice but how general are they?

Author Response

  • The article unfolds on two levels that remain quite separate throughout the text: the economic model and its mathematical analysis. Moreover, I find the treatment of both levels lacking.

We thank the referee for this helpful observation that the economic model and its mathematical analysis were originally “quite separate” and independently lacking. On the mathematical front, we have generalized our model to accommodate the study of any labor economy, rather than focusing on our generated payoffs. We have also reworked the manuscript to explicitly connect core concepts (attractor arc, trapping zones, environment actuated oscillating dynamics, escape) to potential “economic” implications for the labor market.

We achieve both by adding two new sections (see section 3.2, p. 10-15, lines 337-403; and section 3.5, p. 20-22, lines 545-617) and by generalizing the technology (section 3.3, p. 15-17, lines 404-486) and policy (section 3.4, p. 17-20, lines 487-544) sections.

  • On the economic side, the important part is the derivation of the interaction between firms are workers as a battle of sexes. The details, given in appendix A, rely on behavioral rules which I find rather arbitrary. The chosen mathematical models fit better with other social problems, the easiest being coordination on the same action/norm in a multi-cultural society.

We thank the referee for this comment. As we have now generalized our model to all payoffs, we have removed the payoff generation section originally in appendix A from the primary manuscript; in doing so, we shift our focus from the “behavioral rules” used in our payoff generation and instead look towards new insights from the generalized model. Rather than analyzing our generated payoffs, we refocus our analysis to study the new generalized model and explore how the three described environment factors (technology, policy and markets) can work in concert to maintain a pseudo-stable state, i.e., a co-existence of gig and employee strategies. We believe that this major repositioning of our paper clarifies how the chosen mathematical models are appropriate in studying the interaction between firms and laborers in the gig economy. We present our economic insights from the general model in section 3.5.2 (p. 22, lines 576-617).

  • On the mathematical side, the paper is, potentially, more appealing. Letting the payoff structure of the game change (albeit exogenously) is quite interesting. However, I could not find formal results, e.g. about convergence to an arc. Pictures look nice but how general are they?

We thank the referee for this positive comment, and we greatly appreciate this inquiry for general results. In our revision, we have taken steps to generalize our model, enabling the study of any labor economy. In the revised manuscript, we have shifted our findings away from analysis of generated specific payoff values and refocused on insights derived from the new generalized model.  In the revised work, the presented findings are generalizable to all payoffs. We achieve this by adding two new sections (see section 3.2, p. 10-15, lines 337-403; and section 3.5, p. 20-22, lines 545-617) and by generalizing the technology (section 3.3, p. 15-17, lines 404-486) and policy (section 3.4, p. 17-20, lines 487-544) sections.

In terms of formal results about convergence to an arc, we clarify that in the present work, we do not claim an eternal pseudo-stable state, but rather, our three environment-driven dynamics enable persistence in a pseudo-stable state for extended time intervals; there remains the possibility of escape over a long enough period of time. In particular, we have explicitly added nullclines of the systems at market conditions (n=1 and n=0), which fully characterize the regions of pseudo-stable states admitting such oscillations arising from market condition changes. We discuss these dynamics and implications in the newly added section (section 3.5, p. 20-22, lines 545-617).

Reviewer 4 Report

The authors present an evolutionary game theory model for workers choosing between gig economy jobs and regular jobs and companies choosing between gig employment and regular employment. 

The authors highlight that their model is highly stylised. In its simplest form, a firm and a worker meet, and choose whether they propose gig employment or regular employment.

I am a huge fan of stylised models. A simple model can often provide more precise insights. Unfortunately, the framework provided by the authors hides more than it conceals. The authors fail to explain how their model's central concepts (e.g. attractor arc, trapping zones, environment-actuated driven oscillation, and escape) map to labor markets. After reading the paper, the paper left me with the feeling of "so what?". I failed to get the insights as the jump from authors' assumptions to results seems relatively short.

Below are some specific comments on each section. 

Comments on the intro
The authors seem to assume that before platform mediated gig work, the gig workers were mostly worked in  "elite, skilled roles". E.g. "an angel investor or company advisor or perhaps even a Mckinsey consultant typified the variety of early gig positions." 

I disagree with this claim. Day labor was a common source of income for non-skilled workers without access to traditional labour markets. Moreover, the term "gig" refers to musicians and performing artists who hardly compare to consultants. 

As a minor point, Burtch et al is missing from references. 

Comments on the basic model: 
The timing of the basic model is not entirely clear: do the agents know the state of the world (bull vs. bear market) when they meet and choose their strategy or not? I would assume that a worker and an employer would know whether they are in a bear or bull world when they negotiate their terms of employment, but this is not clear from the text.

In some sense, it would make sense that the world would not be flipping between bull and bear markets randomly but that there would be some persistence. How would that affect the results? 

Moran processes are not entirely familiar for someone not well-versed in evolutionary game theory, and they should be explained. 

Turning to results: 
The authors assume that workers have a better bargaining position in a bull market, which seems tenous (as evidenced by payoff matrices in Fig 1).  It is also not clear why firms would prefer hiring employees in a bear market. I would have assumed the opposite: that firms prefer a flexible workforce in the downturn? These assumptions should be much better explained in the text and not buried in the appendices.

I suppose that the payoffs in Fig1 are specific for some firm size/skill level combination, but what? 

Firm types and policy, and technology are possibly helpful extensions to the basic model. Still, I had a hard time seeing what implications they have on the primary results. The strategy mix dynamics seem slightly different, but this should be highlighted more. The authors present an evolutionary game theory model for workers choosing between gig economy jobs and regular jobs and companies choosing between gig employment and regular employment. 

The authors highlight that their model is highly stylised. In its simplest form, a firm and a worker meet, and choose whether they propose gig employment or regular employment.

I am a huge fan of stylised models. A simple model can often provide more precise insights. Unfortunately, the framework provided by the authors hides more than it conceals. The authors fail to explain how their model's central concepts (e.g. attractor arc, trapping zones, environment-actuated driven oscillation, and escape) map to labor markets. After reading the paper, the paper left me with the feeling of "so what?". I failed to get the insights as the jump from authors' assumptions to results seems relatively short.

Below are some specific comments on each section. 

Comments on the intro
The authors seem to assume that before platform mediated gig work, the gig workers were mostly worked in  "elite, skilled roles". E.g. "an angel investor or company advisor or perhaps even a Mckinsey consultant typified the variety of early gig positions." 

I disagree with this claim. Day labor was a common source of income for non-skilled workers without access to traditional labour markets. Moreover, the term "gig" refers to musicians and performing artists who hardly compare to consultants. 

As a minor point, Burtch et al is missing from references. 

Comments on the basic model: 
The timing of the basic model is not entirely clear: do the agents know the state of the world (bull vs. bear market) when they meet and choose their strategy or not? I would assume that a worker and an employer would know whether they are in a bear or bull world when they negotiate their terms of employment, but this is not clear from the text.

In some sense, it would make sense that the world would not be flipping between bull and bear markets randomly but that there would be some persistence. How would that affect the results? 

Moran processes are not entirely familiar for someone not well-versed in evolutionary game theory, and they should be explained. 

Turning to results: 
The authors assume that workers have a better bargaining position in a bull market, which seems tenous (as evidenced by payoff matrices in Fig 1).  It is also not clear why firms would prefer hiring employees in a bear market. I would have assumed the opposite: that firms prefer a flexible workforce in the downturn? These assumptions should be much better explained in the text and not buried in the appendices.

I suppose that the payoffs in Fig1 are specific for some firm size/skill level combination, but what? 

Firm types are possibly helpful extensions to the basic model. Still, I had a hard time seeing what implications they have on the primary results. The strategy mix dynamics seem slightly different, but this should be highlighted more. Could figures 9-12 showing strategy proportions be superimposed? 

The same applies to tech and policy (even though the values seem slightly ad hoc).

Despite the lukewarm comments, I do see the value of the research agenda that the authors have pursued, and I wish them good luck for the future. 

Author Response

  • The authors present an evolutionary game theory model for workers choosing between gig economy jobs and regular jobs and companies choosing between gig employment and regular employment. 

The authors highlight that their model is highly stylised. In its simplest form, a firm and a worker meet, and choose whether they propose gig employment or regular employment.

I am a huge fan of stylised models. A simple model can often provide more precise insights. Unfortunately, the framework provided by the authors hides more than it conceals. The authors fail to explain how their model's central concepts (e.g. attractor arc, trapping zones, environment-actuated driven oscillation, and escape) map to labor markets. After reading the paper, the paper left me with the feeling of "so what?". I failed to get the insights as the jump from authors' assumptions to results seems relatively short.

We thank the referee for their careful reading and their summary of our work. To simplify the manuscript and distill clarity, we have eliminated the computational payoff generation section, originally in appendix A, from the primary manuscript. Additionally, we have generalized the model to enable the study of all labor economies and repositioned our focus towards an analysis of the generalized treble of dynamics (oscillatory dynamics, arc drift, arc tilt). By removing the computational aspect of payoff generation, we have simplified the manuscript and believe that the focus on the new generalized model presents clearer and more precise insights.

Regarding implications for the labor economy, we added a new section (3.5) to bring together all the core concepts in relation to each other and the labor market. We discuss how market, technology and policy environments symbiotically evolve in concert to maintain a pseudo-stable state of gig and employee strategy co-existence. Further, we explore how a new radical policy, technology or market development might be able to introduce imbalance and shift specific labor economies (such as that of a specific industry) to  a single strategy workforce.

We achieve this by adding two new sections (see section 3.2, p. 10-15, lines 337-403; and section 3.5, p. 20-22, lines 545-617) and by generalizing the technology (section 3.3, p. 15-17, lines 404-486) and policy (section 3.4, p. 17-20, lines 487-544) sections.

We thank the referee for their detailed comments, which helped us reposition the paper in an impactful way.

Below are some specific comments on each section. 

Comments on the intro

  • The authors seem to assume that before platform mediated gig work, the gig workers were mostly worked in  "elite, skilled roles". E.g. "an angel investor or company advisor or perhaps even a Mckinsey consultant typified the variety of early gig positions." 

I disagree with this claim. Day labor was a common source of income for non-skilled workers without access to traditional labour markets. Moreover, the term "gig" refers to musicians and performing artists who hardly compare to consultants. 

We apologize for the confusion that we may have caused. Indeed, the term “gig” originally referenced musicians and performing artists but the modern use of the term, “gig worker”, now encompasses a broader workforce (independent contractors, contract workers, freelancers, etc.). We clarify our use of the term “gig labor” in reference to other perspectives and the history of the moniker (p. 16, lines 434-438). With this clarification of roles encompassing “gig work”, we maintain our position that gig labor in the pre-digital era was primarily composed of skilled labor (those who were most able to find and secure gig opportunities without the aid of digital platforms) and that technology enabled more workers to discover gig opportunities and participate as a gig laborer.

“We note that while the term "gig worker" was originally coined in reference to musicians and performing artists \cite{abraham2017measuring}, we apply the contemporary definition of "gig worker" which generalizes to independent contractors, freelancers and temporary contract workers. Therefore, we interpret the roles of certain entrepreneurs, service consultants and corporate advisors as gig labor.”

  • As a minor point, Burtch et al is missing from references. 

We thank the referee for the careful reading of our paper. We have corrected this error.

Comments on the basic model: 

  • The timing of the basic model is not entirely clear: do the agents know the state of the world (bull vs. bear market) when they meet and choose their strategy or not? I would assume that a worker and an employer would know whether they are in a bear or bull world when they negotiate their terms of employment, but this is not clear from the text.

We apologize for the lack of context and for any confusion we may have caused. We have added additional context to explain how agents “learn” the state of the world as modeled through the social learning process (p. 4, lines 178-185).

“When firms and laborers choose to participate in the labor market (i.e., firms hiring for and laborers seeking new employment roles), gig and employee decisions are determined based on respective payoff incentives, and the composition of employees and gig workers evolves accordingly. Firms and laborers will only interact in the labor market when they recognize evolving environments that influence existing strategy payoffs. Here, selection intensity can also be understood as the rate at which firms and laborers realize external factors that cause them to shift strategy preferences. In evolutionary game theory, this social learning process can be modeled as the Moran process \citep{traulsen2009stochastic, de2019fixation}.”

When we introduce our environment variable, n, we also clarify how each player comes to realize the changing environment (p. 5, lines 203-205).

“Having introduced this environment coefficient, selection intensity $\omega$, can be understood as the social learning rate at which firms and laborers acknowledge the economic landscape and realize new payoffs for each strategy.”

  • In some sense, it would make sense that the world would not be flipping between bull and bear markets randomly but that there would be some persistence. How would that affect the results? 

We thank the referee for the comment and clarify that our selected $\dot{n}$ used in demonstrations does persist in each market state (bear and bull market) before transitioning (p. 8, lines 266-275). We also note that the selected $\dot{n}$ is used only in demonstrations and that it can take any function (results in the revised manuscript will not be affected as we primarily focus on the directional dynamics of market change. We clarify the generalized directional dynamics in Figure 10 on p.14 from the newly added section 3.2).

“For our demonstrations, we apply a simple step-wise function for $\dot{n}$ such that the environment instantaneously alternates between $n=0$ and $n=1$ every 5 time units, see Figure A2 in Appendix A.1.3. Regarding the economy, this implies that a bull market will persist for 5 time units before transitioning to a bear market which will also persist for 5 time units. For clarity, we plot our selected $\dot{n}$ to help visualize the rate of change for the environment. Notably, our step-wise $\dot{n}$ implies that the attractor will jump from the two extremes of the attractor arc corresponding to $n=0$ and $n=1$. While we provide a reference attractor arc in all demonstrations, our $\dot{n}$ implies the attractor will not take an intermediary position on the arc. The authors note that the selected $\dot{n}$ is solely used in demonstrations to help illustrate each dynamic in a simple manner. In reality, $\dot{n}$ can take any function, which we discuss further in later sections.”

  • Moran processes are not entirely familiar for someone not well-versed in evolutionary game theory, and they should be explained. 

We thank the reviewer for their thoughtful consideration for how our manuscript reads to broader audiences. We have further described how the Moran process proceeds (p. 4-5, lines 176-191)

“Selection intensity, denoted with $\omega \in [0,1]$, represents the frequency in which firms and laborers interact in the labor market. When firms and laborers do not interact in the labor market, the composition of employees and gig workers remains constant. When firms and laborers choose to participate in the labor market (i.e., firms hiring for and laborers seeking new employment roles), gig and employee decisions are determined based on respective payoff incentives, and the composition of employees and gig workers evolves accordingly. Firms and laborers will only interact in the labor market when they recognize evolving environments that influence existing strategy payoffs. Here, selection intensity can also be understood as the rate at which firms and laborers realize external factors that cause them to shift strategy preferences. In evolutionary game theory, this social learning process can be modeled as the Moran process \citep{traulsen2009stochastic, de2019fixation}. The Moran process proceeds as a stochastic process where one labor contract ends and a new labor contract arises to reflect the latest firm and laborer preferences, which are modeled by the fitness of each strategy type. In our model, $\omega$ in essence constitutes the rate of change for strategy densities in firm and laborer populations. For $\omega$ = 0, the fitness of the strategy type is 0 as the player does not interact in the labor market, and the rate of change for gig-employee strategy densities is 0. When $\omega = 1$, the fitness $f$ equates to the payoff $\pi$ for the strategy type, and firms and laborers engage in the labor market at the maximum cadence. We have

\begin{equation}

f = 1-\omega +{\omega}\pi

\end{equation}”

Turning to results: 

  • The authors assume that workers have a better bargaining position in a bull market, which seems tenous (as evidenced by payoff matrices in Fig 1).  It is also not clear why firms would prefer hiring employees in a bear market. I would have assumed the opposite: that firms prefer a flexible workforce in the downturn? These assumptions should be much better explained in the text and not buried in the appendices.

We apologize for the confusion we may have caused in our theoretical demonstrations. We now clarify that the payoffs in Figure 1 are used solely to introduce and demonstrate our key concepts. (p. 6, lines 219-223)

“We note that the selected theoretical payoff matrices are used in demonstrations only and have no relationships to any specific labor economy. In future sections, we generalize the model to all payoffs. For simplification purposes, we assign all mismatching strategies a payoff of $0$ as no mutual labor agreement is made between firm and laborer. “

Since we have now focused our analysis on the new generalized model, we have removed the mentioned assumptions (part of the payoff generation originally in appendix A) from the primary manuscript.

  • I suppose that the payoffs in Fig1 are specific for some firm size/skill level combination, but what? 

Again, we apologize for the confusion we may have caused in the theoretical demonstrations and clarify that Figure 1 payoffs are used for demonstrative purposes only and do not represent any specific labor economy. (p. 6, lines 219-223)

“We note that the selected theoretical payoff matrices are used in demonstrations only and have no relationships to any specific labor economy. In future sections, we generalize the model to all payoffs. For simplification purposes, we assign all mismatching strategies a payoff of $0$ as no mutual labor agreement is made between firm and laborer. “

  • Firm types are possibly helpful extensions to the basic model. Still, I had a hard time seeing what implications they have on the primary results. The strategy mix dynamics seem slightly different, but this should be highlighted more. Could figures 9-12 showing strategy proportions be superimposed? 

The same applies to tech and policy (even though the values seem slightly ad hoc).

We thank the referee for this comment. We apologize for the lack of clarity in connecting each dynamic (as influenced by market, technology, policy) with primary results and implications for the gig economy. We have generalized the policy (section 3.4, p. 17-20, lines 487-544) , technology (section 3.3, p. 15-17, lines 404-486) and market (see section 3.2, p. 10-15, lines 337-403) sections to be indiscriminate of the provided payoff. Additionally, in section 3.5 (p. 20-22, lines 545-617), we have connected all concepts by demonstrating the three dynamics evolving in concert and discuss implications for the labor market.

The strategy-mix dynamics as referenced in figures 9-12 (in the original manuscript) were based on our generated payoffs (originally in Appendix A). As we have now shifted our focus to insights founded on the new generalized model, our primary discussion moves away from the analysis of firm types.

We believe that the new focus of the primary manuscript clearly depicts the influence that the treble of dynamics has on the labor market. In reference to figures 9-12 (in the original manuscript), which we have now moved to the Supplementary Information file as an example, we chose to keep the figures independent to illustrate each firm-type’s market-driven directional dynamics. We appreciate the referee’s thoughtful inquiry which helped us tie core concepts with implications for primary results in the revised manuscript.

  • Despite the lukewarm comments, I do see the value of the research agenda that the authors have pursued, and I wish them good luck for the future. 

We thank the referee for their thoughtful encouragement and time in reviewing our work.

Reviewer 5 Report

This manuscripts presents the development of a novel model of labor markets. There are two players (i.e., the labor and the firm) representing the probability of agents to exhibit contractual preference for 'gig' work (as opposed to traditional employment). 

The foundations of the model lie in game theory, specifically the replicator equation of evolutionary game theory, a nonlinear differential equation akin to the Lotka-Volterra model (aka, preditor-prey model) where each strategy (i.e, gig or employment) is a species. 

The model is reduced to two coupled first-order differential equations (one for the probability labor prefers gig work, the second for the probability that firms prefer gig work) with three parameters: the evolutionary or social learning rate ($\omega \in [0,1]$), and two 2x2 matrices ($A$ and $B$) that represent the pay offs of a given strategy (probability) for labor and firms respectively. 

The authors further simplify the model to only two scalar parameters ($\omega$) and $n \in [0,1]$, the second being the weighting in the weighted average between bear and bull labor markets and their associated $A$ and $B$. Two stable fixed points are identified, a labor-firm consensus on gig work, and a consensus on employee status. 

The replicator equation model implies that if a strategy ever 'dies' it cannot be 'resurrected'; and since the authors note that gig and employee status have both existed in labor markets in recent history, then the real labor market must not be at equilibrium and instead 'trapped' in an orbit of forced dynamics. The authors explain these forced dynamics as exogenous changes to $\omega$ and $n$. 

This work aims to contribute a needed theoretical bases to inform and support the extensive qualitative and statistical analysis of gig work in the labor market. The paper covers a lot of ground, from changes in markets, policy, and technology. However, leaps and practical claims are made about the model that are not clearly founded in mathematics (i.e., not to say they are invalid claims, but that they are not clearly proven) and contrary points in the literature are not provided (e.g., the increase in gig work such as grocery delivery during the pandemic contrary to the claimed expected decrease in gig work during an economic downturn). 

Regarding mathematical claims and foundations, the authors use the term "trapping zones" which does not seem to have a consensus definition in the literature. If one exists, provide a citation, if one doesn't exist, then clearly define. The authors casually define it in the last paragraph of page 9 with the shepherd and sheep metaphor. 

However, this description does not align with the phase portraits in Figure 7 where the "sheep" could "fall off" the "mountain ridgeline" that the shepherd travels, causing the sheep to descend to one of the stable equilibriums at (1,1) or (0,0). Mathematically, the saddle point moving along the yellow arc in Figure 7 would only be able to bound the system state if the system starts exactly along that arc. Any initial or dynamic deviation along that arc would cause system to no longer be affected by the location of the saddle point since the yellow arc is only marginally stable.

Figures 9(c), 10(c), 11(c), and 12(c) all show a what appears to be "trapping" behavior. However, upon closer inspection it is clearly apparent that the states are converging towards (1,1) in Figure 9 and 10 and diverging in Figure 11 and 12. It leaves the reader wondering if the system would leave the apparent "trap" and settle in a stable equilibrium if enough time was simulated. 

Escape behavior is clearly shown in Figure A2 and A3. Yet, the authors leave the calculation of escape boundary, and thus the existence of stable trapping region, to future work. It is plausible that if the saddle point moves in 2D orbit, like that illustrated in Figure 15, that the system response could be bounded; however, this is not mathematically proven in this manuscript. Without such proof (e.g., based on a Lyapunov equation), the claims on policy, technology, and economics are mere conjecture based on an assumption that a proof exists. 

This reviewer recommends that the authors consider splitting this work into at least two or three manuscripts: one proving the mathematical existence of stable regions under periodic exogenous inputs (affecting $\dot{A}$ and $\dot{B}$) and one or two manuscripts describing the economic, policy, and technical implications of such trapping behavior, and escape, in the gig economy. 

Specific comments and recommendations are listed below:

    • The abstract should summarize key results.
    • Pg 2. "vast multiplex of considerations" -- reword to make more meaningful. 
    • Throughout the manuscript, references to bear and bull markets are mentioned; are these in reference to the national economy, or the labor economy of interest? I.e., micro or macro market?
    • Equations and their context should be added before (4) that better explain the jump between $\pi_i$ in (1), its definition in (4), and the apparent contradiction in (2) that implies $\pi_i = (A \overrightarrow{y})_i$.
    • The introduction is excessively long, recommend reducing from the beginning to "in recent years" by 50%. 
    • When introducing (1)…(6), it should be shown which variables may be time varying by added "(t)" after each time varying variable. 
    • Figure 2: Label $x1$ "labor", $y1$ "firm" and 0.0 "employee" and 1.0 "gig. 
    • Delete Figure 3, 5, 6, 7, unnecessary. 
    • Some readers may be confused around the use of the words market(place), firm, and platform. For example in the case of ride-share, some might consider the driver to be the worker, and the rider to be the one hiring the driver (i.e., that firm), leaving the app (e.g., Uber/Lyft) to be the platform operators. The authors view is not incorrect, but should be clarified with regard to other potential views.
    • The paper is excessively long. See comment above regarding splitting the manuscript. 

Author Response

  • This manuscripts presents the development of a novel model of labor markets. There are two players (i.e., the labor and the firm) representing the probability of agents to exhibit contractual preference for 'gig' work (as opposed to traditional employment). 

The foundations of the model lie in game theory, specifically the replicator equation of evolutionary game theory, a nonlinear differential equation akin to the Lotka-Volterra model (aka, preditor-prey model) where each strategy (i.e, gig or employment) is a species. 

The model is reduced to two coupled first-order differential equations (one for the probability labor prefers gig work, the second for the probability that firms prefer gig work) with three parameters: the evolutionary or social learning rate ($\omega \in [0,1]$), and two 2x2 matrices ($A$ and $B$) that represent the pay offs of a given strategy (probability) for labor and firms respectively. 

The authors further simplify the model to only two scalar parameters ($\omega$) and $n \in [0,1]$, the second being the weighting in the weighted average between bear and bull labor markets and their associated $A$ and $B$. Two stable fixed points are identified, a labor-firm consensus on gig work, and a consensus on employee status. 

The replicator equation model implies that if a strategy ever 'dies' it cannot be 'resurrected'; and since the authors note that gig and employee status have both existed in labor markets in recent history, then the real labor market must not be at equilibrium and instead 'trapped' in an orbit of forced dynamics. The authors explain these forced dynamics as exogenous changes to $\omega$ and $n$. 

We thank the reviewer for their careful reading and accurate summary of our paper.

  • This work aims to contribute a needed theoretical bases to inform and support the extensive qualitative and statistical analysis of gig work in the labor market. The paper covers a lot of ground, from changes in markets, policy, and technology. However, leaps and practical claims are made about the model that are not clearly founded in mathematics (i.e., not to say they are invalid claims, but that they are not clearly proven) and contrary points in the literature are not provided (e.g., the increase in gig work such as grocery delivery during the pandemic contrary to the claimed expected decrease in gig work during an economic downturn). 

We thank the referee for this comment. In the revision, we clarify our claims, generalize our mathematical model and reposition our central analysis towards insights derived from the new generalized model.

In the revised manuscript, we clarify that the trapping zone is a region where the system can persist in a pseudo-stable state for numerous periods, but not necessarily forever. In this sense, the possibility for a labor economy to escape to a single strategy is possible. However, in the revised work, we describe a treble of dynamics that works in concert to evolve the trapping zone and sustain a pseudo-stable state.

Second, we generalize our model to be indiscriminate of the provided payoffs. We reposition the focus of our paper to centralize on insights from this generalized model rather than the analysis of our generated payoffs for specific firm types (which has been removed from the main manuscript). Our findings now focus on how technology, policy and markets evolve to sustain the pseudo-stable state.

  • Regarding mathematical claims and foundations, the authors use the term "trapping zones" which does not seem to have a consensus definition in the literature. If one exists, provide a citation, if one doesn't exist, then clearly define. The authors casually define it in the last paragraph of page 9 with the shepherd and sheep metaphor. 

We apologize for the lack of clarity in the introduction of the trapping zone concept. We now provide a formal definition when we introduce the concept. (p. 9, lines 276-293), and also add nullclines of the systems at market conditions (n=0 and n=1), which can fully characterize the trapping zone of such pseudo-stable states in the oscillatory dynamics.

“For a given pair of GameStates, $\dot{n}$ determines the orbit and moving speed of the attractor. As the attractor orbits the attractor arc, the attractor's oscillation can drive the system to oscillate as well. We refer to this as a driven oscillation. Near the attractor arc, there exists a trapping zone. We introduce the trapping zone as a region where the system can persist in a pseudo-stable state for numerous cycles of environment change. Here, the attractor has a shepherding role. In order for the attractor to herd the system for numerous periods, $\omega\neq0$ must be small enough compared to $\dot{n}\neq0$ such that the system does not escape the ends of the attractor arc. A simple analogy can help elucidate this concept. The attractor behaves as a shepherd who can only move along one line, the attractor arc. The system behaves like a sheep that is running towards or away from the shepherd, depending on the orientation of the attractor arc. The shepherding attractor must move from one end of the arc to the other faster than the sheep in order to trap it. If the sheep reaches an escape boundary such that the shepherding attractor can not keep up, it will escape and end up at one of the two stable equilibria at $(0,0)^*$ or $(1,1)^*$. Escape from the trapping zone depends on the nontrivial relationship between $\dot{n}$ and $\omega$. Therefore, given $\omega$ is very small such that the system is evolving much slower than the attractor, the trapping zone behaves as a pseudo-stable equilibrium between a pair of GameStates. Without environmental changes, the system remains stationary at the attractor arc, and it will slowly escape to one of the stable equilibria.”

  • However, this description does not align with the phase portraits in Figure 7 where the "sheep" could "fall off" the "mountain ridgeline" that the shepherd travels, causing the sheep to descend to one of the stable equilibriums at (1,1) or (0,0). Mathematically, the saddle point moving along the yellow arc in Figure 7 would only be able to bound the system state if the system starts exactly along that arc. Any initial or dynamic deviation along that arc would cause system to no longer be affected by the location of the saddle point since the yellow arc is only marginally stable.

Figures 9(c), 10(c), 11(c), and 12(c) all show a what appears to be "trapping" behavior. However, upon closer inspection it is clearly apparent that the states are converging towards (1,1) in Figure 9 and 10 and diverging in Figure 11 and 12. It leaves the reader wondering if the system would leave the apparent "trap" and settle in a stable equilibrium if enough time was simulated. 

Escape behavior is clearly shown in Figure A2 and A3. Yet, the authors leave the calculation of escape boundary, and thus the existence of stable trapping region, to future work. It is plausible that if the saddle point moves in 2D orbit, like that illustrated in Figure 15, that the system response could be bounded; however, this is not mathematically proven in this manuscript. Without such proof (e.g., based on a Lyapunov equation), the claims on policy, technology, and economics are mere conjecture based on an assumption that a proof exists. 

We thank the referee for these accurate descriptions of the escape and trapping zone concepts we propose. In the section 3.5 of the revised manuscript (p. 20-22, lines 454 – 618), we formalize how the attractor arc (and the trapping zone which follows the arc) moves in 2D orbit to continually re-capture and trap the “sheep” that is descending the “mountain ridgeline”.  

In the revised manuscript, we clarify that the system can persist in the trapping zone for many market cycles (from the genesis of the labor market to the present day given the observable co-existence of labor strategies today), but do not claim that the system can exist eternally in this pseudo-stable state (see the nullclines in Fig. 4 and Fig. 5c in the revised manuscript). Rather, we describe the treble of dynamics as a “robust” balancing act where market, technology and policy environments evolve in concert to sustain a pseudo-stable state for long time intervals (we assume a finite time interval from the genesis of labor markets to the present day). However, in the revised manuscript, we do not claim that the system will remain in this pseudo-stable state forever. In fact, we state that under certain technology, policy or market events, some labor economies (such as that of a specific industry or firm) may experience a dramatic paradigm shift in the nature of their work and shift entirely to a single strategy. In sum, we describe a treble of dynamics that decreases the velocity of escape, but does not necessarily immortalize the system in an eternal trapping zone.

Given that we welcome the possibility of escape and do not claim an eternal pseudo-stable state in the revised manuscript, we concluded that a proof for an eternal trapping zone would be inappropriate in the present work. Given our assumption that the system may only be temporarily “trapped” in a pseudo-stable state (and that it has been for many cycles of environment change), the claims on policy, technology and markets are able to stand alone without the proof of an eternal pseudo-stable equilibrium. The focus of our revised manuscript is not on an “eternal” pseudo-stable equilibrium, and thus, such a proof would be better situated in another work.

  • This reviewer recommends that the authors consider splitting this work into at least two or three manuscripts: one proving the mathematical existence of stable regions under periodic exogenous inputs (affecting $\dot{A}$ and $\dot{B}$) and one or two manuscripts describing the economic, policy, and technical implications of such trapping behavior, and escape, in the gig economy. 

With the reviewer’s comment in mind, we have shortened the manuscript by 25 percent. We believe that the greatly-condensed revised manuscript should not be divided further, but agree that further mathematical analysis (such as that describing an eternal trapping zone) and additional economic analysis should exist as separate works.

Specific comments and recommendations are listed below:

  • The abstract should summarize key results.

We have rewritten the abstract to summarize key results. (p. 1, lines 11-26)

“The emergence of the modern gig economy introduces a new set of employment considerations for firms and laborers that include various trade-offs. With a game-theoretical approach, we examine the influences of technology, policy and markets on firm and worker preferences for gig labor. Theoretically, we present new conceptual extensions to the replicator equation and model oscillating dynamics in two-player asymmetric bi-matrix games with time-evolving environments, introducing concepts of the attractor arc, trapping zone and escape. While canonical applications of evolutionary game theory focus on the evolutionary stable strategy, our model assumes that the system exhibits oscillatory dynamics and can persist for long temporal intervals in a pseudo-stable state. We demonstrate how changing market conditions result in distinct evolutionary patterns across labor economies. Informing tensions regarding the future of this new employment category, we present a novel payoff framework to analyze the role of technology on the growth of the gig economy. Regarding governance, we explore regulatory implications within the gig economy, demonstrating how intervals of lenient and strict policy alter firm and worker sensitivities between gig and employee labor strategies. Finally, we establish an aggregate economic framework to explain how technology, policy and market environments engage in an interlocking dance, a balancing act, to sustain the observable co-existence of gig and employee labor strategies.”

  • Pg 2. "vast multiplex of considerations" -- reword to make more meaningful. 

We now have reworded the sentence to read as follows (p. 2, lines 47-49): “For each labor economy, there exists a crossroads of competing considerations for firms and laborers regarding their labor decisions.”

  • Throughout the manuscript, references to bear and bull markets are mentioned; are these in

reference to the national economy, or the labor economy of interest? I.e., micro or macro market?

We have now added additional clarification that our model can be generalized to any labor economy, whether it is that of a country or a single firm.

“Our model involves a pair of GameStates. GameState pairs consist of a labor economy, whether that of a country or of a single firm, in a bear and bull market.” (p. 5, lines 195-196)

“Therefore, some labor economies, whether it be that of a specific sector or a single firm, may eventually break out of the pseudo-stable trapping zone and escape to an ESS where either only gig or only employee strategies exist.” (p. 24, lines 606-608)

  • Equations and their context should be added before (4) that better explain the jump between $\pi_i$ in (1), its definition in (4), and the apparent contradiction in (2) that implies $\pi_i = (A \overrightarrow{y})_i$.

We have clarified our description of the Moran process, and the text now reads as follows. (p. 2,3, lines 176-191)

“Selection intensity, denoted with $\omega \in [0,1]$, represents the frequency in which firms and laborers interact in the labor market. When firms and laborers do not interact in the labor market, the composition of employees and gig workers remains constant. When firms and laborers choose to participate in the labor market (i.e., firms hiring for and laborers seeking new employment roles), gig and employee decisions are determined based on respective payoff incentives, and the composition of employees and gig workers evolves accordingly. Firms and laborers will only interact in the labor market when they recognize evolving environments that influence existing strategy payoffs. Here, selection intensity can also be understood as the rate at which firms and laborers realize external factors that cause them to shift strategy preferences. In evolutionary game theory, this social learning process can be modeled as the Moran process \citep{traulsen2009stochastic, de2019fixation}. The Moran process proceeds as a stochastic process where one labor contract ends and a new labor contract arises to reflect the latest firm and laborer preferences, which are modeled by the fitness of each strategy type. In our model, $\omega$ constitutes the rate of change for strategy densities in firm and laborer populations. For $\omega$ = 0, the fitness of the strategy type is 0 as the player does not interact in the labor market, and the rate of change for gig-employee strategy densities is 0. When $\omega = 1$, the fitness $f$ equates to the payoff $\pi$ for the strategy type, and firms and laborers engage in the labor market at the maximum cadence. We have

f = 1-\omega +{\omega}\pi”

  • The introduction is excessively long, recommend reducing from the beginning to "in recent years" by 50%. 

We have shortened the introduction as the referee suggested. 

  • When introducing (1)…(6), it should be shown which variables may be time varying by added "(t)" after each time varying variable. 

When we discuss equations (1) … (6), we have not yet introduced our evolving environment variable which is a function of time. Therefore, we decided that it would be inappropriate to mention that some variables are time varying at this point in the derivation. However, once we introduce our environment variable, we state which variables are time varying to address the referee’s request. (p. 5, lines 200-207)

“An environment coefficient, $n \in [0, 1]$, represents market condition. $n = 0$ denotes the bear market and $n = 1$ denotes the bull market. $n$ can take any value

between 0 and 1; for instance, $n = 0.5$ signifies that the environment is a neutral

market, the midway point in a transition between bear and bull market conditions. Having introduced this environment coefficient, selection intensity $\omega$, can be understood as the social learning rate at which firms and laborers acknowledge the economic landscape and realize new payoffs for each strategy. In our model, payoffs $A$ and $B$ are functions of time, but selection intensity $\omega$ is a constant. Applying this environment coefficient, we rephrase our firm and laborer payoffs to account for the domain of market conditions.”

  • Figure 2: Label $x1$ "labor", $y1$ "firm" and 0.0 "employee" and 1.0 "gig. 
  •  

We have now added this additional information in the figure description (Figure 2, p. 7). Figure 2 caption now reads:

“Saddle Point Geographies with Theoretical GameState Payoffs, See Figure 1. (a) Bear GameState $n=0$ (b) Bull GameState $n=1$ (c) Quadrant Legend. $x1$ and $y1$ denote laborer and firm preference for gig work respectively where 1.0 represents a universal gig strategy and 0.0 represents a universal employee strategy.”  (page 2, Figure 2)

  • Delete Figure 3, 5, 6, 7, unnecessary. 

To accommodate readers who are not familiar with evolutionary game theory, we have decided to keep the figures in the manuscript. With the referee’s suggestion in mind, we moved figures 3 and 6 to the appendix for reference. However, we decided to keep figures 5 and 7 in the main text as we believe these are helpful visual aids to assist readers in their understanding of the attractor arc and driven oscillation concepts when they are introduced.

  • Some readers may be confused around the use of the words market(place), firm, and platform. For example in the case of ride-share, some might consider the driver to be the worker, and the rider to be the one hiring the driver (i.e., that firm), leaving the app (e.g., Uber/Lyft) to be the platform operators. The authors view is not incorrect, but should be clarified with regard to other potential views.

We have added additional clarification to address contrasting perspectives. (p. 16,17, lines 442-444)

“While some readers may regard sharing-economy applications as platform operations, drivers as gig workers and customers as the firm who employs the driver, we clarify our interpretation of the platform as the firm.”

  • The paper is excessively long. See comment above regarding splitting the manuscript. 

With the reviewer’s comment in mind, we have shortened the manuscript by 25 percent. We believe that the greatly-condensed revised manuscript should not be divided, but agree that further mathematical analysis (such as that describing an eternal trapping zone) and additional economic analysis should exist as separate works.

We thank the referee for their thoughtful comments and careful reading of our work. We believe these comments have helped the authors improve the revised manuscript in a significant way.